# Zero-shot Benchmarking: A Framework for Flexible and Scalable Automatic Evaluation of Language Models

**José Pombal**[1,2,3], **Nuno M. Guerreiro**[1,2,3,4], **Ricardo Rei**[1] **& André F.T. Martins**[1,2,3,5]

[1]Unbabel, [2]Instituto de Telecomunicações
[3]Instituto Superior Técnico, Universidade de Lisboa
[4]MICS, CentraleSupélec, Université Paris-Saclay, [5]ELLIS Unit Lisbon
pombal.josemaria@gmail.com

## Abstract

As language models improve and grow capable of performing more complex tasks across modalities, evaluating them automatically becomes increasingly challenging. Developing strong and robust task-specific automatic metrics gets harder, and human-annotated test sets—which are expensive to create—saturate more quickly. A compelling alternative is to design reliable strategies to automate the creation of test data and evaluation, but previous attempts either rely on pre-existing data, or focus solely on individual tasks. We present Zero-shot Benchmarking (ZSB), a framework for creating high-quality benchmarks for any task by leveraging language models for both synthetic test data creation and evaluation. ZSB is simple and flexible: it requires only the creation of a prompt for data generation and one for evaluation; it is scalable to tasks and languages where collecting real-world data is costly or impractical; it is model-agnostic, allowing the creation of increasingly challenging benchmarks as models improve. To assess the effectiveness of our framework, we create benchmarks for five text-only tasks and a multi-modal one: general capabilities in four languages (English, Chinese, French, and Korean), translation, and general vision-language capabilities in English. We then rank a broad range of open and closed systems on our benchmarks. ZSB rankings consistently correlate strongly with human rankings, outperforming widely-adopted standard benchmarks. Through ablations, we find that strong benchmarks can be created with open models, and that judge model size and dataset variety are crucial drivers of performance. We release all our benchmarks, and code to reproduce our experiments and to produce new benchmarks.[1]

## 1 Introduction

Automatic evaluation has played a crucial part in the rapid development of large language models (LLMs), as a proxy for more expensive and time-consuming human evaluations. The typical approach to automatic evaluation is to create a human-annotated test set, and then evaluate a model on it using some automatic metric that may be intrinsic (e.g., accuracy on a multiple-choice test), or extrinsic (e.g., BLEURT (Sellam et al., 2020) for text generation). However, as the capabilities of models improve and expand to new tasks and modalities, this approach becomes challenging. Not only because developing accurate and robust metrics for every task is hard, but also because exisiting human-annotated test sets saturate more quickly and are expensive and time-consuming to create.

To address the scarcity of automatic metrics, it has become more common to use LLMs-as-judges (Zheng et al., 2023), wherein an LLM is used to evaluate long-form outputs of systems (Li et al., 2024d; Gao et al., 2024). On the other hand, to tackle the scarcity of test

---

[1]All data, prompts, and code we used and to create and run new benchmarks is on Github.

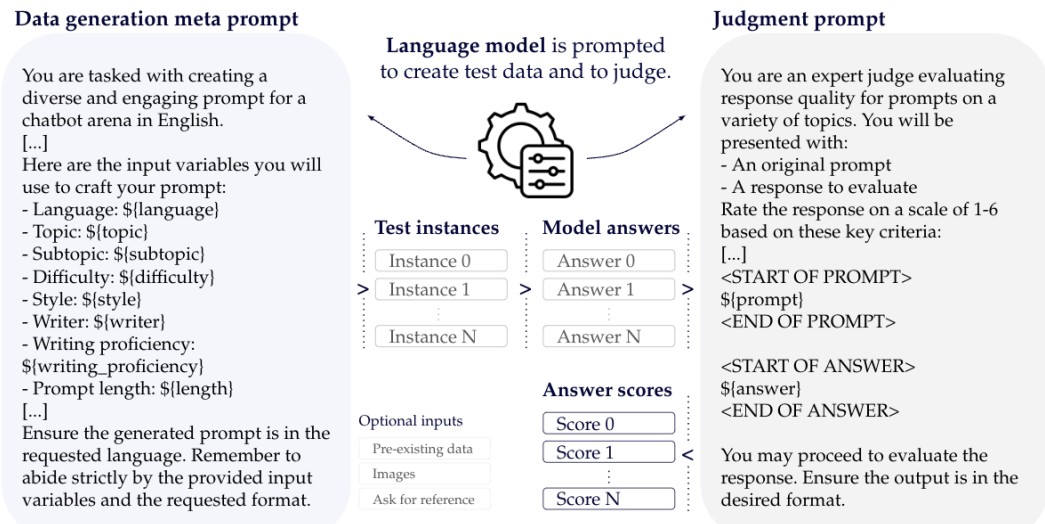

Figure 1: Zero-shot Benchmarking (ZSB) framework and task example. The variables in the meta prompt (inside curly brackets) allow for generating varied test instances. The judgment prompt is flexible to either direct assessment or pairwise evaluation. The same LLM, or different ones may be used for data generation and evaluation.

sets, there have been mainly two trends: 1) popular synthetically-generated multiple-choice benchmarks have emerged, such as MuSR (Sprague et al., 2023) or BoardGameQA (Kazemi et al., 2024); 2) efforts like the Chatbot Arena (Zheng et al., 2023)—a widely-adopted benchmark for language model capabilities—have proposed crowdsourcing the creation of test data and evaluations. Despite the widespread adoption of the latter approach, its limited scalability—due to the reliance on humans—constrains its usability for a wide range of tasks and for model development. Some works propose automating the creation of evaluation data, the evaluation itself, or both. However, they either rely on pre-existing data (Li et al., 2024c), or focus solely on evaluating chat capabilities in English (Zhao et al., 2024; Hu et al., 2025), or summarization (Hu et al., 2025), leaving out most real-world use-cases and tasks.

In this work, we present Zero-shot Benchmarking (ZSB), a framework for creating high-quality benchmarks for any task by leveraging language models for both synthetic test data creation and evaluation. Our framework, showcased in Figure 1, requires as little human effort as the creation of two prompts: *i*) a meta prompt for data generation; and *ii*) a judgment prompt for evaluation with a Likert scale. ZSB unifies two highly impactful language model applications—LLM-as-a-judge and synthetic data generation—into a flexible and scalable framework for end-to-end automatic benchmarking.

ZSB represents a shift in how we approach benchmarking, particularly for specialized or emerging tasks. Unlike traditional benchmarks that require extensive human-annotated data, ZSB can be rapidly deployed for applications where collecting real-world data would be too costly or impractical, like evaluating chat capabilities in non-English languages. ZSB is also flexible and modular, allowing users to seamlessly create benchmarks that evaluate several skills at once, like vision-language chat capabilities, which combine textual and visual understanding. Furthermore, contrary to static test sets, and because the framework is model-agnostic, the quality of Zero-shot benchmarks evolves alongside improvements in LLM capabilities: as more powerful models emerge, both the data generation and evaluation components can be updated simply by switching to newer models, ensuring benchmarks remain relevant and challenging.

To assess the effectiveness of ZSB, we create benchmarks for five text tasks and a multi-modal one: general capabilities[2] in four languages (English, Chinese, French, and Korean), trans-

---

[2]We define "general capabilities" in Section 3.2.

lation (across eleven language pairs), and vision-language general capabilities in English. We evaluate up to 22 LLMs and VLMs on our benchmarks, yielding rankings that correlate strongly with human rankings, often outperforming widely-adopted standard benchmarks. We also perform a series of ablations to show that good benchmarks can be created with open models, and that judge model size and dataset variety are crucial factors for guaranteeing the best performance. Accompanying this work, we release all our benchmarks, as well as code to reproduce our experiments and to produce new benchmarks.[3]

## 2 Zero-shot Benchmarking (ZSB)

ZSB (Figure 1) is a framework for creating benchmarks fully automatically with language models, from test data generation to evaluation, for any task.

**Data generation.** Data generation requires creating a **meta prompt**. The meta prompt specifies the task and requests the generation of an instance that abides by a series of placeholder attributes that are relevant to the task. For example, for chat capabilities in English, our meta prompt contains placeholders for topic, subtopic, difficulty, style, length, and writer background and proficiency, yielding a total of 9,832,320 possible combinations. We include all attributes and their prevalence on all the released benchmarks in Appendix A.2. We later show that variety in placeholders is important for the quality of the benchmark (§5). The meta prompt may also contain a request for a reference answer, or be accompanied by pre-existing data or by images. In order to obtain a useful data instance, it is also required to define a function for parsing the output of the meta prompt. All meta prompts used, as well as examples of generated instances are included in Appendices A.1 and A.4, respectively.

**Evaluation.** Evaluation requires creating a **judgment prompt** that can contain a reference answer. Contrary to previous works, we opt for a direct assessment (Zheng et al., 2023; Liu et al., 2023; Ye et al., 2023; Kim et al., 2023; Graham et al., 2018; 2015, DA) prompt with a 6-point Likert scale, instead of a pairwise comparison prompt (Zheng et al., 2023; Zhao et al., 2024; Luo et al., 2024, PWC). Though DAs pose the challenge of defining an appropriate scoring scale, they are more versatile than PWC evaluation in that they do not require multiple rounds of comparisons to obtain an Elo ranking (Zheng et al., 2023), or the definition of a baseline answer, as done by Li et al. (2024c). Furthermore, a DA prompt can leverage pre-established scoring systems, such as the MQM framework for translation (Lommel et al., 2014; 2013). We show that both DA and PWC yield system rankings of similar quality (§5). All judgment prompts used are included in Appendix A.3. In this work, we only leverage a single model for evaluation,[4] but we note that ZSB can be coupled with human-in-the-loop evaluation, or approaches that attempt to mitigate the biases of a single judge, like querying multiple judges (Verga et al., 2024; Zhao et al., 2024).

**Additional instance-level metadata.** Despite the increasing usage of synthetic benchmarks, a documentation framework is still missing, such as datasheets for datasets (Gebru et al., 2021). In this spirit, we provide a description and a 6-point safety score generated with Claude-Sonnet-3.5 for every instance in our benchmarks. We include the prompt we used for generating safety metadata, safety score distributions for every ZSB task, and three examples in Appendix A.5. The majority of instances has score greater or equal to 5 (Generally Safe), with less than 0.3% receiving scores lower or equal to 3 (Somewhat Risky).

## 3 Experimental Setup

For each task, we create 500 test instances using Claude-Sonnet-3.5[5] (totaling 8000 instances) and evaluate up to 22 systems. We choose a wide variety of open and closed systems

---

[3]Datasets and code will be released upon publication.

[4]We test and compare several different open and closed LLMs as judges (§5).

[5]We choose Claude-Sonnet-3.5 because it is a state-of-the-art LLM, but we try other models (§5).

of different sizes and families; the complete list of evaluated systems per task, as well as their standings in the gold and ZSB rankings, can be found in Appendix B.1. Evaluation is performed with Claude-Sonnet-3.5 using a 6-point Likert scale in the prompt. Systems are ranked according to their average score across all instances.

## 3.1 Meta-evaluation

Benchmarks are usually used to measure progress in the field by ranking competing systems. As such, we evaluate ZSB by comparing its system rankings to well-established human rankings and baselines. We use Pairwise-Accuracy (Kocmi et al., 2021, PA) to measure ranking correlation, which is defined as the fraction of pairs of systems for which the benchmark ranking agrees with the gold ranking. PA is equivalent to the widely-used Kendall $\tau$ rank correlation coefficient (Kendall, 1938) modulo a linear scaling and shifting (Thompson et al., 2024), so it will rank benchmarks in the same order as $\tau$. We prefer to use PA for its intuitive interpretation, i.e., "for any two systems, how likely is the benchmark to rank them in the same order as the gold ranking?". Following previous work (Zheng et al., 2023; Zhao et al., 2024), we include Spearman's $\rho$ in Appendix C.1 (conclusions are the same as with PA).

## 3.2 Tasks

**LLM General Capabilities.** The term "general capabilities" is often used to refer to the real-world utility of language models in addressing queries that involve core knowledge, instruction-following, and conversational capabilities (Zheng et al., 2023). However, traditional test sets often rely on multiple-choice assessments, which have inherent limitations, such as lacking diversity and complexity. The Chatbot Arena (Zheng et al., 2023) offers an alternative approach by crowdsourcing open-ended queries and pairwise response evaluations. Its main advantage is that it is closer to a deployment of an LLM: humans will create diverse and complex queries as they would in a real-world setting. Additionally, by relying on human evaluation, its rankings will reflect human preferences more than multiple-choice accuracy. With over 1 million queries and evaluations, it has become the *de facto* standard for general capabilities evaluation. With ZSB, we want to mimic this approach, but fully automatically. As such, we generate open-ended queries across English, Chinese, French, and Korean and evaluate up to 22 LLMs, comparing our resulting rankings with the Arena's. We include a brief per-category analysis of the ArenaHard data and our final datasets in Appendix D.

**Translation.** Translation is a challenging task because it is cross-lingual and open-ended , and has been at the core of language model development in recent years (e.g., it motivated the Transformer architecture (Vaswani et al., 2017)). Furthermore, it has a wide range of automatic evaluation metrics available that correlate well with human judgments (Freitag et al., 2024), which constitute strong baselines. This provides a unique, challenging testbed for ZSB. Thus, we create benchmarks for translation across 11 language pairs,[6] and evaluate up to 7 systems. We use the WMT24 General MT Shared task (Kocmi et al., 2024) standings as gold rankings. The WMT shared task is a major annual competition in the field.[7] Each year, professional translators are hired to create high-quality test sets for a variety of language pairs. Shared task participants then submit translations, which professional translators evaluate. Given this reliance on professionals for test set creation and evaluation, the resulting rankings are considered the standard for comparing MT system quality.

**VLM English General Capabilities.** General capabilities here extend beyond text to include visual understanding, where Vision-Language Models (VLMs) have recently shown impressive performance (Zhang et al., 2024). However, automatic evaluation for VLM tasks is underdeveloped compared to its text-only counterpart: high-quality benchmarks are scarce and evaluation frameworks are harder to set up and use. Thus, it is an excellent

---

[6]English-German, -Czech, -Spanish, -Russian, -Japanese, -Chinese, -Hindi, -Ukrainian, -Icelandic, Czech-Ukrainian, and Japanese-Chinese.
[7]With over fifteen editions, WMT has become one of the biggest events at *ACL conferences.

candidate to test ZSB and its extendibility to different modalities. We evaluate a total of 12 VLMs, and leverage gold rankings from the vision section of the Chatbot Arena.

### 3.3 Baselines

**Standard Benchmarks.** We adopt widely-used standard benchmarks as baselines (we list all benchmarks used in Appendix B). For LLM and VLM general capabilities, these benchmarks are a combination of static multiple-choice test sets. We derive correlations with gold rankings in three different ways, which depend on how we obtain system rankings: i) by averaging the scores of each system on each test set, and then ranking systems (Average, the most common approach); ii) by taking the Borda Count (Colombo et al., 2022), where we average the ranking of each system on each benchmark (Borda); iii) by only considering the test set in the benchmark with the highest correlations with gold rankings (Top-1). Our goal is to establish a challenging baseline by considering the multiple ways in which practitioners might interpret benchmark results. Note that the Borda count value for each system depends on the number of systems being considered, and that Top-1 is an oracle baseline (i.e., we know which benchmark correlates best with the gold ranking when selecting it).

The setup is slightly different for translation. Although test sets are also static, task-specific automatic metrics are used to evaluate systems automatically. As such, we consider the system rankings on WMT24 data yielded by the metric used by the WMT24 organizers, AUTORANK (Kocmi et al., 2024)[8], and three state-of-the-art metrics: reference-based xCOMET-XXL (Guerreiro et al., 2023) and METRICX-24 (Juraska et al., 2024), and reference-free COMETKIWI-XXL (Rei et al., 2023). These baselines are challenging, as they leverage test data created by professional translators—which also underpins the gold rankings—as well as strong, well-established MT-specific automatic metrics for evaluation.

**M-ArenaHard Upper Bound.** For LLM general capabilities, we consider an additional baseline: M-ArenaHard (Dang et al., 2024), which was obtained by translating ArenaHard (Li et al., 2024c) with Google Translate. ArenaHard contains 500 challenging Chatbot Arena queries in English. We use our judging framework and not the original implementation to isolate the impact of the test data's provenance on ranking performance. We consider this to be an upper bound on the performance of our benchmarks, as the queries of ArenaHard were taken directly from the source of the gold rankings we use.

## 4 Main Results

### 4.1 LLM General Capabilities

**ZSB outperforms standard benchmarks across all languages and is competitive with test sets obtained by curating human data.** Table 1 shows the PA scores for each language. ZSB outperforms averaging test sets in standard benchmarks—one of the most used method for evaluating general capabilities—across the board. Furthermore, it outperforms Borda count— the strongest baseline—on French and Korean. The same trend is observed when comparing against almost all individual benchmarks, instead of aggregations (see Appendix B.2). Remarkably, ZSB outperforms the upper bound in French at 0.845 PA. These findings outline the potential of ZSB for creating high-quality multilingual benchmarks at a low cost: our English test set cost roughly $5, while ArenaHard (Li et al., 2024c) cost $500, and GPQA (Rein et al., 2023)—which got 0.73 PA—cost upward of $100,000 (Zheng et al., 2023).

### 4.2 Translation

**ZSB is competitive with combining human test data and MT-specific automatic metrics.** Table 2 shows PA scores for MT, aggregated across language pairs (we show by-LP results

---

[8]To get system-level scores, AUTORANK linearly scales and averages the system-level scores (an average over the scores of all translations) of two state-of-the-art MT metrics: METRICX-23-XL (Juraska et al., 2024) (reference-based), and COMETKIWI-XL (Rei et al., 2023) (reference-free).

| | LLM General Capabilities | | | | VLM General Capabilities |
| | English | Chinese | French | Korean | English |
|---|---|---|---|---|---|
| **Baselines** | | | | | |
| M-ArenaHard | *0.9048* | *0.9004* | 0.8333 | *0.9231* | - |
| Std. Benchmarks (Average) | 0.8268 | 0.8269 | 0.7576 | 0.7949 | 0.8182 |
| Std. Benchmarks (Borda) | **0.8874** | **0.8528** | 0.7273 | 0.8205 | 0.8182 |
| Std. Benchmarks (Top-1) | 0.8485 | 0.8268 | 0.7576 | 0.8718 | 0.8636 |
| **ZSB (ours)** | 0.8571 | 0.8355 | *0.8485* | **0.8462** | **0.8636** |

Table 1: Pairwise Accuracy (PA) scores for LLM and VLM general capabilities across languages. Bold denotes the best score apart from the upper bound (M-ArenaHard), and italics denote the best score overall. Top-1 baseline is greyed out because it is an oracle.

| | WMT24 Test Data | | | | ZSB (ours) |
| | METRICX-24-XXL | XCOMET-XXL | COMETKIWI-XXL | AUTORANK | LLM Judge |
|---|---|---|---|---|---|
| **Translation** | 0.8322 | 0.8322 | 0.8112 | 0.8182 | 0.7902 |

Table 2: Pairwise Accuracy (PA) scores for translation. We consider up to 7 systems across 11 language pairs, totalling 143 system comparisons.

in Appendix C.5). At 0.79 PA, ZSB is only two points below AUTORANK and COMETKIWI-XXL, the state-of-the-art reference-free metric for MT. ZSB is also close to reference-based XCOMET and METRICX. This is a remarkable result given the complexity of the task and the quality of the baseline: not only are the metrics we report tailored for MT, but the data underlying their system rankings is the same as the gold rankings. Because of the high cost of hiring professionals to create data and evaluate systems, WMT lacks language coverage and many flavours of translation (e.g., transcreation, which mixes translation with rewriting for cultural adaptation). ZSB could potentially be used for creating benchmarks for any language and task (or mixture thereof) with minimal human effort, which may alleviate this issue.

### 4.3 VLM English General Capabilities

**Data creation.** For each instance of this task, we randomly sample an image—and only the image—from the validation set of `textvqa` (Singh et al., 2019), a question-answering dataset with a wide variety of images. The meta prompt then queries the model to generate a prompt related to the image in the same style of the LLM general capabilities task, but without attribute placeholders. An interesting direction for future work would be creating a test set from synthetically-generated images.

**ZSB outperforms multimodal standard benchmarks.** Table 1 shows that ZSB outperforms standard benchmarks by a considerable margin—4.5 PA points—regardless of the aggregation method. This is a strong result, considering that the leaderboard leverages 8 datasets, while our dataset contains only 500 instances. Notably, all but one test set, MMMU, are outperformed by ZSB (see Table 23 in the Appendix). This highlights the flexibility of Zero-shot Benchmarking in seamlessly extending to multimodal settings.

| | | | | Judge | | | |
|---|---|---|---|---|---|---|---|
| **Data Generator** | Claude | Llama-70B | Qwen-3B | Qwen-7B | Qwen-14B | Qwen-32B | Qwen-72B |
| Claude | *0.8528*\* | 0.8182 | **0.7446** | 0.7013 | 0.7100 | 0.8139 | 0.7792 |
| Llama-70B | 0.7532 | *0.7792* | 0.6277 | 0.6797 | 0.7273 | 0.7706 | 0.7619 |
| Qwen-3B | 0.7706 | *0.7792* | 0.6840 | 0.6623 | 0.7403 | 0.7749 | 0.7662 |
| Qwen-7B | *0.7835* | 0.7749 | 0.6753 | 0.6667 | 0.6970 | 0.7576 | 0.7489 |
| Qwen-14B | *0.8052* | 0.7922 | 0.7143 | **0.7273** | 0.7446 | 0.7749 | 0.7965 |
| Qwen-32B | *0.7922* | *0.7922* | 0.6537 | 0.7100 | **0.7922** | 0.7619 | 0.8139 |
| Qwen-72B | 0.7922 | *0.8268* | 0.6797 | 0.7100 | 0.7446 | **0.8312** | **0.8182** |

Table 3: Pairwise accuracy of Zero-shot Benchmark for general capabilities in English with various data generation and judge models. Italic denotes the best judge for a given generator; bold the best data generator for a given judge. The \* denotes our original configuration.

## 5  Ablations

We ablate a series of components in our framework and experimental setup to understand their impact on the quality of the ZSB text-only general capabilities benchmarks.

**Open models can be used to generate high-quality benchmark data.**   The vast majority benchmarks comprised of synthetic data were generated with proprietary models (Fan et al., 2023; Sprague et al., 2023; Wu et al., 2024). Using open models for data generation is an attractive alternative for three reasons: *i)* data may be cheaper to generate; *ii)* the resulting data can be free of usage restrictions; and, perhaps most importantly, *iii)* it is easier to improve benchmarks as ZSB improves in tandem with open model advancements. Thus, we evaluate the performance of open models Llama-3.3-70B-Instruct (Grattafiori et al., 2024) and Qwen2.5-{3, 7, 14, 32, 72}B-Instruct (Yang et al., 2024b) on both the data generation and judgment components of our framework. Table 3 shows that benchmarks created with Llama-3.3-70B-Instruct or Qwen2.5-72B-Instruct can achieve strong correlations with human rankings in English, and, remarkably, outside of English (see Appendix C.2). Importantly, Qwen outperforms averaging standard benchmarks across the board. Choosing the right model is crucial for the quality of the benchmark—for example, Qwen is especially strong in Korean, while Llama lags behind. Finally, to further illustrate that the quality of ZSB can improve alongside that of open models, we present results for general capabilities using Llama 4 Maverick (Llama 4 Team, 2025) as a data generator and judge in Appendix C.4, showing it outperforms Llama 3.3 on 3 out of 4 languages.

**Open models are strong judges but they must be somewhat large.**   Table 3 also shows the performance of different judge models, given the same test data. Though smaller Qwen models perform worse in general, high performance can be achieved from 14B parameters onwards, especially for Chinese and Korean (see Appendix C.2). Notably, using Llama-3.3-70B-Instruct as a judge with data generated by Claude-Sonnet-3.5 leads to better ranking correlations on Chinese and French, even outperforming M-ArenaHard on the latter (see Appendix C.2). It is also viable to use open models on the full ZSB stack, i.e., both data generation and evaluation: pairing Qwen and Llama, or Qwen models models can lead to higher PA outside of English than exclusively using Claude.

**Judges perform better on data created by models of the same family.**   An interesting trend from Table 3 is that Qwen judges tend to perform better on data generated by other Qwen models. For example, from sizes 7B to 72B, Qwen judges often perform better on Qwen-generated data than on data generated by Claude, otherwise the best data generator. A possible explanation is that models are more likely to generate judgments in the desired format when they are used on data generated by their family (see Table 28 in the Appendix).

| | | Claude | | | Llama | |
|---|---|---|---|---|---|---|
| | DA | PWC w/ Baseline | ArenaHard Elo | | DA | PWC All |
| **Pairwise Accuracy** | **0.8528**\* | 0.8268 | 0.8442 | | 0.8355 | 0.8139 |

Table 4: Pairwise-accuracy of ZSB for general capabilities in English with various judgment approaches. The * denotes our original configuration. In "PWC w/ Baseline", the final score of each system is the win rate against GPT-4o. In "ArenaHard Elo", we employ the approach of Li et al. (2024c) to compute an Elo ranking, using Claude-Sonnet-3.5 generations as the baseline answers with PWC evaluation. In "PWC All", we compare all systems against each other, and obtain an Elo ranking from those battles. In the latter, we employ Llama-3.3-70B-Instruct as a judge instead of Claude-Sonnet-3.5 for cost reasons.

While in this work we focused on Claude, this shows that there are performance gains in further tailouring the data generation and judging processes to open models.

**Direct assessment is a viable alternative to pairwise judging.** In this work, we opted for a direct assessment judgment approach instead of a pairwise one. DA makes it easier to add new models to a leaderboard, and can be integrated with existing task-specific scoring frameworks. However, to the best of our knowledge, this type of judging had not been employed before, as past works use forms of PWC to obtain Elo scores for ranking systems (Zheng et al., 2023; Zhao et al., 2024). In Table 4, we show that the performance of DA and PWC is comparable in English, with DA slightly outperforming PWC among all systems, or PWC against a baseline answer generated by Claude-Sonnet-3.5 (the method for computing Elo rankings used in ArenaHardAuto (Zheng et al., 2023)).

**Data instances adhere to meta-prompt specifications, and benchmark performance is robust to meta-prompt variations, but reference answer quality is contentious.** Throughout this work, we only used a single meta-prompt for creating general capabilities data, and we mostly performed reference-less evaluation. However, one step to ensure the practical utility and versatility of our framework is to verify its reliability with respect to the reference answer it provides and to the meta-prompt—both in terms of whether the generated instances respect it, and whether useful instances are still generated after modifying the prompt lexically. Thus, in Table 5 we present five statistics for all the languages we consider: (i) the percentage of data instances that abide by the meta-prompt's attributes; (ii) the percentage of perfect references (scored 1 to 6); (iii) the percentage of score 5 references; (iv) and (v) the PA average and standard deviation of our benchmarks generated with three meta-prompts (the default prompt plus two rewrites).[9] The statistics show similar patterns across languages. While the generated data is consistent with meta prompt specifications, reference quality could be improved: the vast majority of references are quite good (score 5), but only a small fraction are perfect.

**The importance of dataset size is limited beyond 100 instances.** Figure 2a shows the PA scores of our English benchmark over different dataset sizes. While performance increases substantially in the first 100 instances, it stagnates after that, only reaching 2.5 extra PA points at 500 instances. This finding hints at the limited importance of dataset size, but it is encouraging in that small datasets are more accessible and easier to create and curate.

**Dataset variety is a strong driver of benchmark performance.** Figure 2b shows the performance of different iterations of the data generation process, where we vary the dataset size, the number of attributes used in the meta prompt, and the usage of a reference. Adding a reference leads to marginal improvements in performance. Crucially, adding more

---

[9]We prompt Claude-Sonnet-3.5 to obtain all statistics; the scripts and prompts used are in the Github repository linked at the beginning of this work.

| | LLM General Capabilities | | | |
|---|---|---|---|---|
| | English | Chinese | French | Korean |
| **Baselines** | | | | |
| % consistency w/ meta prompt attributes | 95.8 | 97.4 | 99.6 | 99.0 |
| % perfect references | 7.4 | 26.8 | 5.2 | 2.0 |
| % score 5 references | 89.2 | 68.6 | 86.6 | 88.2 |
| % Avg. PA for 3 meta prompts | 0.8470 | 0.8341 | 0.8283 | 0.8419 |
| % Std. dev. of PA for 3 meta prompts | 0.0090 | 0.0025 | 0.0175 | 0.0074 |

Table 5: Data statistics towards ensuring the practical robustness and versatility of our data generation framework: (i) the percentage of data instances that abide by the meta-prompt's attributes; (ii) the percentage of perfect references (scored 1 to 6); (iii) the percentage of score 5 references; (iv) and (v) the PA average and standard deviation of our benchmarks generated with three meta-prompts (the default prompt plus two rewrites).

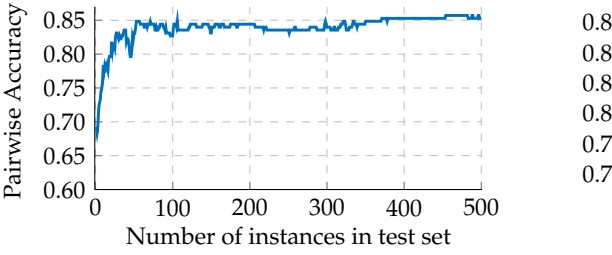
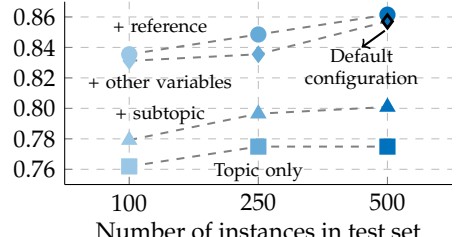

(a) PA for varying dataset sizes at a fine-grained scale.

(b) PA for varying dataset sizes (hue) and meta prompt setups (marker).

Figure 2: Analysis of factors affecting the PA of the ZSB for general capabilities in English.

attribute variety in the meta prompt can lead to improvements of more than 6 PA points. This finding suggests that performance can be further increased with more variety through adding more attributes, or even leveraging external sources of data.

## 6 Related Work

**Fully-automated evaluation suites.** We propose and show the effectiveness of a simple framework to automate test data creation and evaluation for language modeling tasks. Previous work is mainly inspired by the Chatbot Arena (Zheng et al., 2023), where both steps are crowd-sourced. While involving humans enables the collection of gold standard test data and evaluations, it has two shortcomings: 1) evaluation is slow and public, so it is unsuitable for model development; 2) it requires a critical mass of users to be effective, which limits its reach to less popular, yet relevant, tasks. Li et al. (2024c) propose a method for creating synthetic test sets for chatbots by leveraging pre-existing data from the Chatbot Arena. While this solves the first issue, it does not solve the second, since it requires pre-existing data. Zhao et al. (2024) propose automating both data creation and evaluation, but they focus solely on chatbot capabilities in English and Chinese,[10] leaving out other real-world use-cases and tasks. Hu et al. (2025) also focus solely on dialogue response generation and summarization in English. Luo et al. (2024) extend the approach to the video modality and perform in-house meta-evaluation, since there is no established gold ranking. Butt et al. (2024) propose a framework to create benchmarks with several LLMs as agents,

---

[10]They do not compare against other baselines for Chinese, and their reported performance for the Open LLM Leaderboard is low (0.325 Spearman), raising questions about their meta-evaluation setup.

and test seven LLMs on generated calendar scheduling and constrained text generation benchmarks (both in English). ZSB builds on these works by facilitating the exploration of a wider variety of languages and tasks, including multimodal ones. Furthermore, we dissect the impact of different components of our framework on benchmark performance.

**Automatic evaluation with static test sets.** Automatic evaluation is an integral part of language model development. The most common approach is to create a human-annotated test set, and then evaluate the model on it using some automatic metric. Metrics can be intrinsic, like accuracy for multiple-choice tests like MMLU (Hendrycks et al., 2021), and BLEU (Papineni et al., 2002) for MT, or learned models themselves, like COMET (Rei et al., 2020) for MT, or BLEURT (Sellam et al., 2020) for text generation. The judgments of the latter are usually more costly to obtain, but correlate better with human judgments. With the rapid expansion of capabilities in language models, the development of new task-specific metrics has become harder. Thus, LLMs have been increasingly used as judges (Zheng et al., 2023; Gu et al., 2024; Li et al., 2024a;b, LLM-as-a-judge). Some judges, like Prometheus (Kim et al., 2023), are explicitly designed for this purpose through finetuning, while others are simply general-purpose models (Zheng et al., 2023; Kocmi & Federmann, 2023). We employ judge LLMs in our framework, where they excel across all tasks.

**Synthetic test sets.** Another consequence of the increasing capabilities of language models is that test sets saturate faster. To address this challenge, several work introduce synthetically-generated test sets (Wu et al., 2024; Weston et al., 2015; Gandhi et al., 2024; Le et al., 2019; Clark et al., 2020; Saparov & He, 2022; Sinha et al., 2019; Dalvi et al., 2021; Sprague et al., 2022; Kazemi et al., 2024). The advantage of this approach is that it is much more scalable than human-annotated test sets, and they can be enhanced easily as language models improve. One notable is example is MuSR (Sprague et al., 2023), a multiple-choice reasoning benchmark used in the widely-adopted Open LLM Leaderboard. Our framework benefits from the main advantages of synthetic test sets: scalability, and the capacity to create increasingly varied and challenging benchmarks as models improve.

## 7 Conclusion

We presented Zero-shot Benchmarking (ZSB), a framework for creating high-quality benchmarks for language modeling tasks by automating both data creation and evaluation with language models. Through extensive experiments across five text tasks and a multi-modal one, we showed that our framework produces system rankings that correlate well with human judgments, outperforming widely-used standard benchmarks. Our ablation studies revealed several key findings: 1) open models can be used to create high-quality benchmarks, though judge model size is crucial; 2) direct assessments are a viable alternative to pairwise comparisons; 3) dataset variety is more important than size for benchmark quality.

The implications of our work are significant. ZSB enables rapid development of benchmarks for specialized tasks where collecting human-annotated data would be costly or impractical. Furthermore, its flexibility allows benchmarks to evolve alongside improvements in language model capabilities, ensuring continued relevance. We release all our benchmarks and code to facilitate adoption and further research in this direction. In the future we would like to explore applying the framework to other modalities and tasks, investigating methods for synthetic image generation in multimodal benchmarks, and developing techniques to further increase dataset variety (e.g., through the retrieval of relevant context to the task). Additionally, studying the relationship between benchmark performance and real-world utility remains an important open question.

## Acknowledgements

We thank Sweta Agrawal, Giuseppe Attanasio, Duarte Alves, António Farinhas, Haau-Sing Li, and Beatriz Canaverde for their constructive feedback on the paper. We acknowledge EuroHPC JU for awarding the project ID EHPC-AI-2024A01-085 access to MareNostrum 5

ACC. This work was supported by EU's Horizon Europe Research and Innovation Actions (UTTER, contract 101070631), by the project DECOLLAGE (ERC-2022-CoG 101088763), by the Portuguese Recovery and Resilience Plan through project C64500888200000055 (Center for Responsible AI), and by Fundação para a Ciência e Tecnologia through contract UIDB/50008/2020.

## Reproducibility Statement

We release all our benchmarks and code to reproduce our experiments and to create new benchmarks. Part of our experiments rely on closed models, which may become unavailable in the future, posing a potential challenge for reproducibility.

## Ethics Statement

Our work focuses on automating benchmark creation, which raises two key concerns. First, synthetic data may encode biases present in the models used to create it. While we include safety scores for transparency, these are not a complete solution—users should carefully review generated instances before deployment. Second, automated evaluation could be misused to claim superiority without proper validation. We emphasize that our benchmarks should complement, not replace, careful human evaluation and real-world testing. We release our code and data to enable scrutiny and improvement by the research community.

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

# A  Prompts and ZSB Examples

## A.1  ZSB Meta-prompts for Data Generation

You are tasked with creating a diverse and engaging prompt for a chatbot arena. This prompt will be used to test and compare the capabilities of different language models. Your goal is to generate a question or prompt that will challenge these models and showcase their strengths or weaknesses. Also, generate a reference answer to your prompt that will serve as a benchmark for evaluating the models' responses.

Here are the input variables you will use to craft your prompt:
- Language: ${language}
- Topic: ${topic}
- Subtopic: ${subtopic}
- Difficulty: ${difficulty}
- Style: ${style}
- Writer: ${writer}
- Writing proficiency: ${writing_proficiency}
- Prompt length: ${length}

Guidelines for creating the prompt:

1. Abide strictly by the input variables provided.

2. Ensure that your prompt is open-ended enough to allow for varied and interesting responses from different language models.

3. Avoid prompts that are overly specific to a particular AI model's capabilities or training data.

4. Create a prompt that is engaging and thought-provoking, encouraging creative or analytical thinking.

Generate output in the following format:

<START OF PROMPT>
[Your generated prompt here. IMPORTANT: include only the prompt.]
<END OF PROMPT>

<START OF REFERENCE>
[A reference answer to your prompt. IMPORTANT: include only the reference answer.]
<END OF REFERENCE>

Ensure the generated prompt is in the requested language. Remember to abide strictly by the provided input variables and the requested format.

Figure 3: Meta-prompt for multilingual general capabilities.

You are a multilingual content creator and translation expert. Your task is to generate a comprehensive translation exercise based on the given attributes. Follow these instructions carefully:

1. Review the following input variables:
- Source language: ${source_language}
- Target language: ${target_language}
- Topic: ${topic}
- Subtopic: ${subtopic}
- Source Length: ${source_length}
- Style: ${style}

2. Generate a source text: Create an original text in the source language, adhering to the specified topic, subtopic, and length. The text should be coherent, informative, and suitable for translation.

3. Generate a reference translation: Produce a high-quality, fluent translation of the source text in the target language. This translation should serve as a reference for evaluating other translations. IT IS CRUCIAL THAT THE REFERENCE TRANSLATION SOUNDS NATURAL IN THE TARGET LANGUAGE.

Format your output as follows:

<START OF SOURCE>
[INSERT THE SOURCE TEXT HERE]
<END OF SOURCE>

<START OF REFERENCE TRANSLATION>
[INSERT THE REFERENCE TRANSLATION HERE]
<END OF REFERENCE TRANSLATION>

Ensure that your response is comprehensive, coherent, and follows all the instructions provided above. Abide strictly by the requested format and generated until the end of the requested output.

Figure 4: ZSB meta-prompt for translation.

You are tasked with creating a diverse and engaging prompt for a vision language model chatbot arena, given an input image. This prompt will be used to test and compare the capabilities of different vision language models when they analyze images. Your goal is to generate a question or prompt that will challenge these models and showcase their strengths or weaknesses in image understanding, analysis, and reasoning. Also, generate a reference answer that will serve as a benchmark for evaluating the models' responses.

Guidelines for creating the prompt: - Ensure the prompt is relevant to the attached image.
- Create a prompt that requires both visual analysis and reasoning capabilities.
- Do not describe the image in the prompt; let the model analyze it.

Ensure your prompt encourages the model to:
- Describe specific details in the image.
- Make connections between different elements.
- Draw conclusions or insights.
- Provide explanations or interpretations.
- Consider context and implications.

Avoid prompts that:
- Can be answered without looking at the image.
- Focus only on simple object detection.
- Are biased toward specific vision models.

Make prompts that test various vision-language capabilities such as:
- Spatial reasoning
- Object relationships
- Visual detail recognition
- Context understanding
- Abstract concept interpretation
- Temporal reasoning from static images
- Cultural or symbolic understanding

Generate output in the following format:
<START OF PROMPT>
[Your generated prompt here. This should be a question or instruction that will be paired with the image for the model to analyze.]
<END OF PROMPT>

<START OF REFERENCE>
[A reference answer, and only the answer, to the generated prompt.]
<END OF REFERENCE>

Remember to abide strictly by the provided input variables and the requested format.

Figure 5: ZSB meta-prompt for VLM general capabilities. For each generated instance, this prompt is accompanied with some image.

## A.2 Meta-prompt Attributes and Distributions

| Meta-prompt Attribute | LLM General Capabilities | Translation |
|---|---|---|
| **Topic & Subtopic** | | |
| Tech innovation | 1.60% | 1.80% |
| Artificial intelligence | 0.00% | 0.20% |
| Quantum computing | 0.40% | 0.20% |
| Robotics | 0.40% | 0.00% |
| 5g/6g networks | 0.40% | 0.00% |
| Biotechnology | 0.00% | 0.40% |
| Green tech | 0.00% | 0.40% |
| Edge computing | 0.20% | 0.20% |
| Cybersecurity | 0.20% | 0.40% |
| Global markets | 1.80% | 2.00% |
| Stock exchanges | 0.00% | 0.20% |
| Cryptocurrency | 0.20% | 0.60% |
| International trade | 0.60% | 0.20% |
| Foreign investment | 0.20% | 0.00% |
| Commodity markets | 0.20% | 0.40% |
| Emerging markets | 0.40% | 0.20% |
| Foreign exchange | 0.00% | 0.40% |
| Market regulations | 0.60% | 0.00% |
| Environmental policy | 1.20% | 1.00% |
| Carbon trading | 0.20% | 0.20% |
| Renewable energy initiatives | 0.00% | 0.40% |
| Wildlife protection | 0.60% | 0.20% |
| Urban planning | 1.00% | 1.20% |
| Waste management | 0.20% | 0.00% |
| Climate agreements | 0.00% | 0.20% |
| Marine conservation | 0.00% | 0.00% |
| Public health | 1.40% | 1.80% |
| Disease prevention | 0.40% | 0.00% |
| Healthcare systems | 0.40% | 0.20% |
| Vaccination programs | 0.00% | 0.00% |
| Mental health services | 0.40% | 0.80% |
| Maternal health | 0.00% | 0.20% |
| Epidemiology | 0.20% | 0.40% |
| Health technology | 0.00% | 0.20% |
| Urban development | 1.40% | 1.00% |
| Smart cities | 0.20% | 0.20% |
| Public transportation | 0.40% | 0.00% |
| Housing projects | 0.60% | 0.00% |
| Green spaces | 0.00% | 0.40% |
| Infrastructure | 0.20% | 0.40% |
| Urban planning | 1.00% | 1.20% |
| Sustainable development | 0.40% | 0.20% |

Table 6: Part 1 of all meta-prompt attribute values and their prevalence on all Zero-Shots Benchmarks released. Distributions are the same across languages and language pairs, respectively. The VLM task meta-prompt does not have attributes, only an image.

| Meta-prompt Attribute | LLM General Capabilities | Translation |
|---|---|---|
| **Topic** | | |
| International relations | 1.80% | 2.00% |
|   Diplomatic missions | 0.20% | 0.00% |
|   Trade agreements | 0.40% | 0.60% |
|   Cultural exchange | 0.00% | 0.20% |
|   Peace negotiations | 0.20% | 0.00% |
|   International aid | 0.00% | 0.40% |
|   Global security | 0.60% | 0.40% |
|   Regional alliances | 0.40% | 0.40% |
| Education reform | 1.40% | 1.20% |
|   Digital learning | 0.40% | 0.20% |
|   Curriculum changes | 0.20% | 0.00% |
|   Teacher training | 0.20% | 0.40% |
|   Assessment methods | 0.00% | 0.40% |
|   Special education | 0.20% | 0.20% |
|   Higher education | 0.20% | 0.00% |
|   Stem initiatives | 0.20% | 0.00% |
| Cultural trends | 1.20% | 2.20% |
|   Social media influence | 0.00% | 0.40% |
|   Fashion movements | 0.00% | 0.20% |
|   Entertainment trends | 0.00% | 0.40% |
|   Digital culture | 0.40% | 0.20% |
|   Cultural festivals | 0.40% | 1.00% |
|   Art movements | 0.20% | 0.20% |
|   Pop culture | 1.40% | 0.60% |
| Scientific discoveries | 1.60% | 1.00% |
|   Space exploration | 0.20% | 0.00% |
|   Medical breakthroughs | 0.00% | 0.20% |
|   Physics advances | 0.20% | 0.00% |
|   Genetic research | 0.20% | 0.00% |
|   Archaeological finds | 0.60% | 0.20% |
|   Marine biology | 0.20% | 0.00% |
|   Climate science | 0.40% | 0.80% |
| Economic policy | 1.40% | 1.00% |
|   Monetary policy | 0.40% | 0.00% |
|   Fiscal measures | 0.40% | 0.20% |
|   Trade regulations | 0.20% | 0.20% |
|   Tax reform | 0.00% | 0.00% |
|   Employment policy | 0.20% | 0.20% |
|   Banking regulations | 0.00% | 0.00% |
|   Economic stimulus | 0.20% | 0.40% |
| Sports industry | 0.80% | 1.40% |
|   Professional leagues | 0.00% | 0.00% |
|   Sports technology | 0.20% | 0.20% |
|   Athletic training | 0.20% | 0.20% |
|   Sports medicine | 0.00% | 0.60% |
|   E-sports | 0.60% | 0.20% |
|   Sports management | 0.20% | 0.40% |
|   Athletic equipment | 0.00% | 0.00% |
| Media & entertainment | 1.80% | 1.80% |
|   Streaming services | 0.40% | 0.00% |
|   Film production | 0.20% | 0.20% |
|   Gaming industry | 0.40% | 0.00% |
|   Digital media | 0.20% | 0.20% |
|   Publishing | 0.20% | 0.60% |
|   Broadcasting | 0.20% | 0.60% |
|   Content creation | 0.40% | 0.40% |

Table 7: Part 2 of all meta-prompt attribute values and their prevalence on all Zero-Shots Benchmarks released. Distributions are the same across languages and language pairs, respectively. The VLM task meta-prompt does not have attributes, only an image.

| Meta-prompt Attribute | LLM General Capabilities | Translation |
|---|---|---|
| **Topic** | | |
| Workplace transformation | 1.40% | 0.60% |
| Remote work | 0.00% | 0.00% |
| Office technology | 0.60% | 0.00% |
| Employee wellness | 0.20% | 0.00% |
| Corporate culture | 0.00% | 0.20% |
| Hr innovation | 0.20% | 0.00% |
| Workplace safety | 0.20% | 0.20% |
| Professional development | 0.20% | 0.20% |
| Transportation & mobility | 0.40% | 1.40% |
| Electric vehicles | 0.20% | 0.40% |
| Autonomous driving | 0.00% | 0.20% |
| Public transit | 0.00% | 0.40% |
| Aviation | 0.20% | 0.20% |
| Ride sharing | 0.00% | 0.20% |
| Maritime transport | 0.00% | 0.20% |
| Urban mobility | 0.20% | 0.20% |
| Food & agriculture | 1.60% | 0.80% |
| Sustainable farming | 0.40% | 0.00% |
| Food technology | 0.20% | 0.40% |
| Organic production | 0.40% | 0.40% |
| Agricultural policy | 0.00% | 0.00% |
| Food safety | 0.00% | 0.00% |
| Urban farming | 0.00% | 0.20% |
| Agricultural trade | 0.60% | 0.00% |
| Medical & healthcare | 0.80% | 1.00% |
| Telemedicine | 0.20% | 0.00% |
| Medical devices | 0.20% | 0.20% |
| Pharmaceutical research | 0.00% | 0.00% |
| Healthcare it | 0.20% | 0.00% |
| Patient care | 0.00% | 0.40% |
| Medical insurance | 0.20% | 0.40% |
| Clinical trials | 0.40% | 0.00% |
| Legal & compliance | 1.00% | 1.40% |
| Corporate law | 0.00% | 0.20% |
| Intellectual property | 0.00% | 0.20% |
| Data protection | 0.20% | 0.20% |
| Regulatory compliance | 0.00% | 0.60% |
| International law | 0.20% | 0.00% |
| Consumer rights | 0.00% | 0.00% |
| Legal technology | 0.60% | 0.40% |
| E-commerce & retail | 1.40% | 2.00% |
| Online marketplaces | 0.20% | 0.20% |
| Digital payment | 0.40% | 0.20% |
| Retail technology | 0.40% | 0.00% |
| Supply chain | 0.00% | 0.60% |
| Customer experience | 0.00% | 0.00% |
| Mobile commerce | 0.00% | 0.40% |
| Retail analytics | 0.40% | 0.60% |
| Financial services | 1.80% | 1.20% |
| Digital banking | 0.80% | 0.00% |
| Investment management | 0.20% | 0.20% |
| Insurance | 0.40% | 0.20% |
| Payment systems | 0.00% | 0.40% |
| Wealth management | 0.20% | 0.40% |
| Risk management | 0.20% | 0.00% |
| Financial technology | 0.00% | 0.00% |

Table 8: Part 3 of all meta-prompt attribute values and their prevalence on all Zero-Shots Benchmarks released. Distributions are the same across languages and language pairs, respectively. The VLM task meta-prompt does not have attributes, only an image.

| Meta-prompt Attribute | LLM General Capabilities | Translation |
|---|---|---|
| **Topic** | | |
| Gaming & software | 1.40% | 0.80% |
| Game development | 0.40% | 0.00% |
| Cloud gaming | 0.60% | 0.00% |
| Mobile games | 0.40% | 0.00% |
| Gaming hardware | 0.20% | 0.00% |
| Software development | 0.00% | 0.40% |
| Virtual reality | 0.00% | 0.20% |
| Gaming communities | 0.00% | 0.20% |
| Marketing & advertising | 0.60% | 0.60% |
| Digital marketing | 0.20% | 0.20% |
| Brand strategy | 0.00% | 0.20% |
| Social media marketing | 0.20% | 0.20% |
| Content marketing | 0.20% | 0.00% |
| Market research | 0.00% | 0.00% |
| Advertising technology | 0.00% | 0.00% |
| Campaign management | 0.00% | 0.00% |
| Government documentation | 0.80% | 1.40% |
| Policy documents | 0.00% | 0.40% |
| Legal records | 0.00% | 0.20% |
| Public notices | 0.20% | 0.00% |
| Official forms | 0.20% | 0.40% |
| Legislative documents | 0.20% | 0.00% |
| Regulatory guidelines | 0.00% | 0.20% |
| Administrative procedures | 0.20% | 0.20% |
| Academic research | 1.40% | 0.40% |
| Scientific publications | 0.80% | 0.20% |
| Research methodology | 0.20% | 0.00% |
| Peer review | 0.00% | 0.00% |
| Academic collaboration | 0.20% | 0.00% |
| Research funding | 0.00% | 0.20% |
| Data analysis | 0.20% | 0.00% |
| Research ethics | 0.00% | 0.00% |
| Patents & intellectual property | 0.60% | 1.00% |
| Patent applications | 0.00% | 0.20% |
| Trademark registration | 0.00% | 0.00% |
| Copyright protection | 0.00% | 0.40% |
| Ip litigation | 0.20% | 0.00% |
| International patents | 0.20% | 0.00% |
| Trade secrets | 0.00% | 0.20% |
| Ip strategy | 0.20% | 0.20% |
| Manufacturing & safety | 1.00% | 1.20% |
| Quality control | 0.20% | 0.20% |
| Industrial safety | 0.20% | 0.40% |
| Production processes | 0.20% | 0.20% |
| Workplace standards | 0.00% | 0.40% |
| Equipment safety | 0.20% | 0.00% |
| Safety regulations | 0.20% | 0.00% |
| Risk assessment | 0.20% | 0.20% |
| Tourism & hospitality | 0.80% | 1.20% |
| Hotel management | 0.00% | 0.20% |
| Travel services | 0.00% | 0.20% |
| Tourism marketing | 0.20% | 0.20% |
| Cultural tourism | 0.00% | 0.00% |
| Eco-tourism | 0.20% | 0.40% |
| Event planning | 0.20% | 0.00% |
| Customer service | 0.20% | 0.20% |

Table 9: Part 4 of all meta-prompt attribute values and their prevalence on all Zero-Shots Benchmarks released. Distributions are the same across languages and language pairs, respectively. The VLM task meta-prompt does not have attributes, only an image.

| Meta-prompt Attribute | LLM General Capabilities | Translation |
|---|---|---|
| **Topic** | | |
| Religious & cultural studies | 1.40% | 1.00% |
|   Religious traditions | 0.00% | 0.20% |
|   Cultural heritage | 1.60% | 3.20% |
|   Interfaith dialogue | 0.40% | 0.40% |
|   Religious practices | 0.00% | 0.20% |
|   Cultural anthropology | 0.60% | 0.20% |
|   Sacred texts | 0.40% | 0.00% |
|   Religious education | 0.00% | 0.00% |
| Insurance & risk management | 1.20% | 0.80% |
|   Insurance products | 0.40% | 0.20% |
|   Risk assessment | 0.20% | 0.20% |
|   Claims processing | 0.00% | 0.20% |
|   Underwriting | 0.20% | 0.00% |
|   Insurance technology | 0.20% | 0.00% |
|   Regulatory compliance | 0.00% | 0.60% |
|   Risk mitigation | 0.20% | 0.00% |
| Consumer electronics | 1.00% | 1.20% |
|   Mobile devices | 0.00% | 0.20% |
|   Smart home | 0.40% | 0.40% |
|   Wearable technology | 0.00% | 0.40% |
|   Audio equipment | 0.20% | 0.20% |
|   Display technology | 0.20% | 0.00% |
|   Gaming hardware | 0.20% | 0.00% |
|   Personal computing | 0.00% | 0.20% |
| Pharmaceutical industry | 0.80% | 0.80% |
|   Drug development | 0.00% | 0.20% |
|   Clinical trials | 0.40% | 0.00% |
|   Regulatory approval | 0.00% | 0.00% |
|   Manufacturing | 0.20% | 0.40% |
|   Research & development | 0.20% | 0.20% |
|   Market access | 0.00% | 0.00% |
|   Patient safety | 0.00% | 0.00% |
| Fashion & apparel | 1.40% | 1.00% |
|   Fashion design | 0.20% | 0.00% |
|   Textile industry | 0.20% | 0.20% |
|   Sustainable fashion | 0.20% | 0.40% |
|   Retail fashion | 0.00% | 0.00% |
|   Fashion technology | 0.00% | 0.20% |
|   Luxury brands | 0.80% | 0.80% |
|   Fashion marketing | 0.20% | 0.00% |
| Beauty & cosmetics | 0.80% | 1.20% |
|   Skincare | 0.20% | 0.00% |
|   Makeup products | 0.20% | 0.00% |
|   Natural cosmetics | 0.00% | 0.40% |
|   Beauty technology | 0.40% | 0.20% |
|   Product development | 0.00% | 0.40% |
|   Marketing | 0.00% | 0.20% |
|   Sustainability | 0.20% | 0.00% |
| Home & living | 1.00% | 0.60% |
|   Interior design | 0.00% | 0.60% |
|   Smart home | 0.40% | 0.40% |
|   Furniture | 0.40% | 0.00% |
|   Home improvement | 0.20% | 0.20% |
|   Sustainable living | 0.00% | 0.00% |
|   Home technology | 0.20% | 0.00% |
|   Decorative arts | 0.20% | 0.20% |

Table 10: Part 5 of all meta-prompt attribute values and their prevalence on all Zero-Shots Benchmarks released. Distributions are the same across languages and language pairs, respectively. The VLM task meta-prompt does not have attributes, only an image.

| Meta-prompt Attribute | LLM General Capabilities | Translation |
|---|---|---|
| **Topic** | | |
| Automotive industry | 1.20% | 1.00% |
|   Vehicle manufacturing | 0.20% | 0.00% |
|   Electric vehicles | 0.20% | 0.40% |
|   Autonomous technology | 0.40% | 0.40% |
|   Auto design | 0.00% | 0.20% |
|   Car technology | 0.00% | 0.00% |
|   Safety systems | 0.00% | 0.00% |
|   Market trends | 0.40% | 0.00% |
| Social media | 0.60% | 1.80% |
|   Platform development | 0.00% | 0.20% |
|   Content creation | 0.40% | 0.40% |
|   Social analytics | 0.00% | 0.20% |
|   User engagement | 0.00% | 0.40% |
|   Social commerce | 0.00% | 0.00% |
|   Digital communities | 0.00% | 0.00% |
|   Influencer marketing | 0.40% | 0.80% |
| Dating & relationships | 0.40% | 0.60% |
|   Online dating | 0.40% | 0.00% |
|   Relationship counseling | 0.00% | 0.20% |
|   Dating apps | 0.00% | 0.20% |
|   Social connection | 0.00% | 0.00% |
|   Relationship psychology | 0.00% | 0.20% |
|   Dating culture | 0.00% | 0.00% |
|   Personal growth | 0.00% | 0.00% |
| Parenting & family | 2.20% | 0.80% |
|   Child development | 0.00% | 0.00% |
|   Family health | 0.40% | 0.00% |
|   Education | 1.00% | 0.20% |
|   Family dynamics | 0.20% | 0.20% |
|   Parenting resources | 0.20% | 0.00% |
|   Child safety | 0.20% | 0.40% |
|   Family planning | 0.40% | 0.00% |
| Arts & culture | 0.80% | 2.20% |
|   Visual arts | 0.00% | 0.00% |
|   Performance art | 0.00% | 0.60% |
|   Cultural heritage | 1.60% | 3.20% |
|   Art education | 0.00% | 0.20% |
|   Digital art | 0.20% | 0.20% |
|   Cultural events | 0.20% | 0.20% |
|   Art market | 0.00% | 0.20% |
| Music industry | 0.60% | 0.40% |
|   Music production | 0.00% | 0.00% |
|   Digital distribution | 0.20% | 0.20% |
|   Live events | 0.40% | 0.00% |
|   Music technology | 0.00% | 0.00% |
|   Artist management | 0.00% | 0.00% |
|   Music publishing | 0.00% | 0.20% |
|   Industry trends | 0.00% | 0.00% |
| Film & cinema | 1.60% | 1.00% |
|   Film production | 0.20% | 0.20% |
|   Distribution | 0.00% | 0.00% |
|   Film technology | 0.60% | 0.20% |
|   Cinema innovation | 0.00% | 0.00% |
|   Film industry | 0.40% | 1.00% |
|   Digital effects | 0.80% | 0.60% |
|   Film marketing | 0.20% | 0.20% |

Table 11: Part 6 of all meta-prompt attribute values and their prevalence on all Zero-Shots Benchmarks released. Distributions are the same across languages and language pairs, respectively. The VLM task meta-prompt does not have attributes, only an image.

| Meta-prompt Attribute | LLM General Capabilities | Translation |
|---|---|---|
| **Topic** | | |
| Books & literature | 1.20% | 1.00% |
|   Publishing industry | 0.40% | 0.20% |
|   Digital publishing | 0.00% | 0.20% |
|   Literary trends | 0.40% | 0.20% |
|   Book marketing | 0.00% | 0.20% |
|   Author platform | 0.00% | 0.00% |
|   Literary events | 0.40% | 0.20% |
|   Reading technology | 0.00% | 0.00% |
| Food & cuisine | 0.80% | 1.80% |
|   Culinary arts | 1.20% | 0.60% |
|   Food culture | 1.00% | 1.20% |
|   Restaurant industry | 0.00% | 0.60% |
|   Food innovation | 0.00% | 0.20% |
|   Dietary trends | 0.00% | 0.20% |
|   Food technology | 0.20% | 0.40% |
|   Culinary education | 0.40% | 0.20% |
| Sports & recreation | 0.40% | 0.80% |
|   Professional sports | 0.00% | 0.20% |
|   Recreational activities | 0.00% | 0.00% |
|   Sports technology | 0.20% | 0.20% |
|   Fitness training | 0.00% | 0.40% |
|   Sports medicine | 0.00% | 0.60% |
|   Equipment innovation | 0.00% | 0.00% |
|   Sports management | 0.20% | 0.40% |
| Fitness & wellness | 1.00% | 2.00% |
|   Exercise programs | 0.20% | 0.00% |
|   Wellness technology | 0.00% | 0.60% |
|   Mental health | 0.20% | 0.40% |
|   Nutrition | 0.00% | 0.60% |
|   Personal training | 0.40% | 0.00% |
|   Wellness education | 0.20% | 0.20% |
|   Health tracking | 0.00% | 0.20% |
| Mental health | 2.20% | 1.80% |
|   Therapy services | 0.00% | 0.80% |
|   Mental health technology | 0.00% | 0.20% |
|   Support programs | 1.00% | 0.40% |
|   Research & treatment | 0.00% | 0.20% |
|   Workplace mental health | 0.80% | 0.20% |
|   Youth mental health | 0.20% | 0.00% |
|   Mental health education | 0.20% | 0.00% |
| Architecture & design | 0.60% | 1.80% |
|   Urban architecture | 0.20% | 0.00% |
|   Sustainable design | 0.00% | 0.40% |
|   Interior design | 0.00% | 0.60% |
|   Digital architecture | 0.00% | 0.00% |
|   Design innovation | 0.00% | 0.20% |
|   Building technology | 0.20% | 0.00% |
|   Architectural heritage | 0.20% | 0.60% |
| Real estate | 2.20% | 1.80% |
|   Property development | 0.40% | 0.20% |
|   Real estate technology | 0.20% | 0.20% |
|   Market analysis | 0.60% | 0.20% |
|   Property management | 0.20% | 0.00% |
|   Investment | 0.20% | 0.20% |
|   Commercial real estate | 0.20% | 0.40% |
|   Sustainable building | 0.40% | 0.60% |

Table 12: Part 7 of all meta-prompt attribute values and their prevalence on all Zero-Shots Benchmarks released. Distributions are the same across languages and language pairs, respectively. The VLM task meta-prompt does not have attributes, only an image.

| Meta-prompt Attribute | LLM General Capabilities | Translation |
|---|---|---|
| **Topic** | | |
| Telecommunications | 1.60% | 1.20% |
|   Network infrastructure | 0.80% | 0.20% |
|   Mobile technology | 0.20% | 0.20% |
|   Communication services | 0.00% | 0.60% |
|   Digital networks | 0.00% | 0.00% |
|   Telecom innovation | 0.40% | 0.20% |
|   Wireless technology | 0.00% | 0.00% |
|   Industry standards | 0.20% | 0.00% |
| Renewable energy | 1.80% | 1.40% |
|   Solar power | 0.40% | 0.00% |
|   Wind energy | 0.20% | 0.20% |
|   Energy storage | 0.20% | 0.40% |
|   Green technology | 0.20% | 0.20% |
|   Energy policy | 0.40% | 0.20% |
|   Sustainable development | 0.40% | 0.20% |
|   Clean energy innovation | 0.00% | 0.20% |
| Space exploration | 1.60% | 1.20% |
|   Space technology | 0.00% | 0.00% |
|   Satellite systems | 0.00% | 0.20% |
|   Space research | 0.00% | 0.20% |
|   Space industry | 0.20% | 0.00% |
|   Astronomical discovery | 0.80% | 0.40% |
|   Space travel | 0.20% | 0.00% |
|   Space policy | 0.40% | 0.40% |
| Wildlife & nature | 1.20% | 1.80% |
|   Conservation | 0.60% | 0.40% |
|   Biodiversity | 0.00% | 0.20% |
|   Wildlife research | 0.00% | 0.00% |
|   Environmental protection | 0.20% | 0.20% |
|   Natural habitats | 0.40% | 0.20% |
|   Species preservation | 0.00% | 0.40% |
|   Ecosystem management | 0.00% | 0.40% |
| Weather & climate | 1.20% | 1.00% |
|   Climate science | 0.40% | 0.80% |
|   Weather forecasting | 0.00% | 0.40% |
|   Climate change | 0.20% | 0.00% |
|   Atmospheric research | 0.20% | 0.00% |
|   Environmental impact | 0.00% | 0.00% |
|   Climate technology | 0.60% | 0.20% |
|   Weather systems | 0.00% | 0.20% |
| History & heritage | 1.40% | 1.20% |
|   Historical research | 0.20% | 0.40% |
|   Cultural preservation | 0.20% | 0.00% |
|   Archaeological studies | 0.40% | 0.20% |
|   Heritage conservation | 0.20% | 0.20% |
|   Historical education | 0.20% | 0.00% |
|   Digital archives | 0.00% | 0.20% |
|   Cultural memory | 0.20% | 0.20% |
| Politics & governance | 2.20% | 1.20% |
|   Political systems | 0.20% | 0.40% |
|   Public policy | 0.40% | 0.40% |
|   Government innovation | 0.20% | 0.00% |
|   Electoral processes | 0.80% | 0.00% |
|   Political communication | 0.40% | 0.20% |
|   Civic technology | 0.00% | 0.20% |
|   Governance reform | 0.20% | 0.00% |

Table 13: Part 8 of all meta-prompt attribute values and their prevalence on all Zero-Shots Benchmarks released. Distributions are the same across languages and language pairs, respectively. The VLM task meta-prompt does not have attributes, only an image.

| Meta-prompt Attribute | LLM General Capabilities | Translation |
|---|---|---|
| **Topic** | | |
| Ngos & nonprofits | 1.00% | 1.40% |
|   Social impact | 0.00% | 0.00% |
|   Humanitarian aid | 0.40% | 0.40% |
|   Nonprofit management | 0.20% | 0.00% |
|   Fundraising | 0.20% | 0.00% |
|   Community development | 0.00% | 0.20% |
|   International development | 0.00% | 0.40% |
|   Social innovation | 0.20% | 0.40% |
| New york city | 1.20% | 1.40% |
|   Business & finance | 0.00% | 0.40% |
|   Arts & culture | 0.20% | 0.20% |
|   Food & dining | 0.60% | 0.40% |
|   Urban development | 1.40% | 1.40% |
|   Tourism | 1.00% | 0.60% |
|   Entertainment | 0.60% | 0.80% |
|   Sports teams | 0.00% | 0.20% |
| London | 0.60% | 1.80% |
|   Financial services | 0.20% | 0.00% |
|   Cultural heritage | 1.60% | 3.20% |
|   Food scene | 0.80% | 0.40% |
|   Theatre & arts | 0.00% | 0.40% |
|   Royal traditions | 0.40% | 0.00% |
|   Urban transport | 0.00% | 0.00% |
|   Sports culture | 0.60% | 1.00% |
| Tokyo | 1.60% | 0.80% |
|   Technology industry | 0.20% | 0.60% |
|   Traditional culture | 0.40% | 0.40% |
|   Cuisine | 0.00% | 0.00% |
|   Pop culture | 1.40% | 0.60% |
|   Urban innovation | 0.60% | 0.80% |
|   Fashion | 0.40% | 0.40% |
|   Entertainment districts | 0.40% | 0.40% |
| Paris | 1.00% | 1.40% |
|   Fashion industry | 0.20% | 0.60% |
|   Culinary arts | 1.20% | 0.60% |
|   Cultural landmarks | 0.20% | 0.00% |
|   Art scene | 0.20% | 0.20% |
|   Tourism industry | 0.80% | 0.80% |
|   Luxury brands | 0.80% | 0.80% |
|   Urban life | 0.60% | 0.60% |
| Berlin | 2.20% | 1.60% |
|   Startup scene | 0.40% | 0.20% |
|   Cultural history | 0.00% | 0.80% |
|   Nightlife | 0.40% | 0.00% |
|   Art community | 0.40% | 0.00% |
|   Tech industry | 1.20% | 0.80% |
|   Urban planning | 1.00% | 1.20% |
|   Alternative culture | 0.40% | 0.20% |
| Singapore | 1.00% | 1.00% |
|   Financial hub | 0.20% | 0.40% |
|   Food culture | 1.00% | 1.20% |
|   Smart city initiatives | 0.00% | 0.00% |
|   Cultural diversity | 0.00% | 0.20% |
|   Business innovation | 0.20% | 1.20% |
|   Urban planning | 1.00% | 1.20% |
|   Education | 1.00% | 0.20% |

Table 14: Part 9 of all meta-prompt attribute values and their prevalence on all Zero-Shots Benchmarks released. Distributions are the same across languages and language pairs, respectively. The VLM task meta-prompt does not have attributes, only an image.

| Meta-prompt Attribute | LLM General Capabilities | Translation |
|---|---|---|
| **Topic** | | |
| Dubai | 1.60% | 1.00% |
|   Business center | 0.80% | 0.20% |
|   Tourism industry | 0.80% | 0.80% |
|   Luxury lifestyle | 0.20% | 0.00% |
|   Urban development | 1.40% | 1.40% |
|   International trade | 0.60% | 0.20% |
|   Technology innovation | 0.20% | 0.00% |
|   Cultural traditions | 0.60% | 0.40% |
| São paulo | 0.80% | 1.00% |
|   Business hub | 1.20% | 1.40% |
|   Cultural scene | 0.20% | 0.00% |
|   Food & dining | 0.60% | 0.40% |
|   Urban life | 0.60% | 0.60% |
|   Entertainment | 0.60% | 0.80% |
|   Fashion industry | 0.20% | 0.60% |
|   Sports culture | 0.60% | 1.00% |
| Sydney | 1.60% | 0.60% |
|   Lifestyle & culture | 0.20% | 0.20% |
|   Business district | 0.00% | 0.00% |
|   Tourism | 1.00% | 0.60% |
|   Sports events | 0.20% | 0.00% |
|   Entertainment | 0.60% | 0.80% |
|   Food scene | 0.80% | 0.40% |
|   Urban development | 1.40% | 1.40% |
| Mumbai | 1.00% | 2.40% |
|   Film industry | 0.40% | 1.00% |
|   Business center | 0.80% | 0.20% |
|   Cultural heritage | 1.60% | 3.20% |
|   Food culture | 1.00% | 1.20% |
|   Urban development | 1.40% | 1.40% |
|   Entertainment | 0.60% | 0.80% |
|   Fashion | 0.40% | 0.40% |
| Madrid | 1.00% | 1.20% |
|   Cultural heritage | 1.60% | 3.20% |
|   Food & wine | 1.20% | 0.60% |
|   Sports culture | 0.60% | 1.00% |
|   Arts scene | 1.00% | 0.40% |
|   Business hub | 1.20% | 1.40% |
|   Urban life | 0.60% | 0.60% |
|   Tourism | 1.00% | 0.60% |
| Lisbon | 0.80% | 1.80% |
|   Startup ecosystem | 0.20% | 0.00% |
|   Cultural history | 0.00% | 0.80% |
|   Food & wine | 1.20% | 0.60% |
|   Tourism industry | 0.80% | 0.80% |
|   Urban innovation | 0.60% | 0.80% |
|   Maritime heritage | 0.00% | 0.20% |
|   Arts scene | 1.00% | 0.40% |
| Stockholm | 0.80% | 1.40% |
|   Tech innovation | 0.20% | 0.00% |
|   Design culture | 0.20% | 0.00% |
|   Sustainability | 0.20% | 0.00% |
|   Urban planning | 1.00% | 1.20% |
|   Business hub | 1.20% | 1.40% |
|   Cultural scene | 0.20% | 0.00% |
|   Food & lifestyle | 0.00% | 0.20% |

Table 15: Part 10 of all meta-prompt attribute values and their prevalence on all Zero-Shots Benchmarks released. Distributions are the same across languages and language pairs, respectively. The VLM task meta-prompt does not have attributes, only an image.

| Meta-prompt Attribute | LLM General Capabilities | Translation |
|---|---|---|
| **Topic** | | |
| Amsterdam | 1.20% | 0.80% |
|   Cultural heritage | 1.60% | 3.20% |
|   Urban planning | 1.00% | 1.20% |
|   Business innovation | 0.20% | 1.20% |
|   Art scene | 0.20% | 0.20% |
|   Cycling culture | 0.40% | 0.00% |
|   Tourism | 1.00% | 0.60% |
|   Tech industry | 1.20% | 0.80% |
| Seoul | 2.40% | 1.80% |
|   Tech industry | 1.20% | 0.80% |
|   Pop culture | 1.40% | 0.60% |
|   Food scene | 0.80% | 0.40% |
|   Fashion trends | 0.60% | 0.40% |
|   Urban innovation | 0.60% | 0.80% |
|   Entertainment | 0.60% | 0.80% |
|   Business hub | 1.20% | 1.40% |
| Japan | 1.00% | 1.40% |
|   Traditional culture | 0.40% | 0.40% |
|   Technology industry | 0.20% | 0.60% |
|   Popular culture | 0.00% | 0.20% |
|   Business practices | 0.20% | 0.20% |
|   Food & cuisine | 0.40% | 0.20% |
|   Arts & crafts | 0.20% | 0.00% |
|   Social customs | 0.00% | 0.40% |
| France | 1.80% | 1.00% |
|   Culinary arts | 1.20% | 0.60% |
|   Fashion industry | 0.20% | 0.60% |
|   Cultural heritage | 1.60% | 3.20% |
|   Wine industry | 0.40% | 0.20% |
|   Tourism | 1.00% | 0.60% |
|   Business culture | 0.60% | 0.40% |
|   Arts & literature | 0.40% | 0.00% |
| Germany | 1.00% | 1.80% |
|   Automotive industry | 0.40% | 0.20% |
|   Business innovation | 0.20% | 1.20% |
|   Cultural traditions | 0.60% | 0.40% |
|   Sports culture | 0.60% | 1.00% |
|   Technology sector | 0.20% | 0.00% |
|   Education system | 0.20% | 0.20% |
|   Urban development | 1.40% | 1.40% |
| Brazil | 1.20% | 0.80% |
|   Cultural festivals | 0.40% | 1.00% |
|   Sports culture | 0.60% | 1.00% |
|   Business environment | 0.20% | 0.60% |
|   Food & cuisine | 0.40% | 0.20% |
|   Music scene | 0.00% | 0.40% |
|   Tourism industry | 0.80% | 0.80% |
|   Urban life | 0.60% | 0.60% |
| India | 1.40% | 1.20% |
|   Technology sector | 0.20% | 0.00% |
|   Cultural diversity | 0.00% | 0.20% |
|   Film industry | 0.40% | 1.00% |
|   Culinary traditions | 0.40% | 0.00% |
|   Business hub | 1.20% | 1.40% |
|   Traditional arts | 0.00% | 0.20% |
|   Festival culture | 0.00% | 0.20% |

Table 16: Part 11 of all meta-prompt attribute values and their prevalence on all Zero-Shots Benchmarks released. Distributions are the same across languages and language pairs, respectively. The VLM task meta-prompt does not have attributes, only an image.

| Meta-prompt Attribute | LLM General Capabilities | Translation |
|---|---|---|
| **Topic** | | |
| Italy | 1.40% | 0.80% |
|     Food & wine | 1.20% | 0.60% |
|     Fashion industry | 0.20% | 0.60% |
|     Cultural heritage | 1.60% | 3.20% |
|     Tourism sector | 0.20% | 0.40% |
|     Design industry | 0.00% | 0.20% |
|     Business culture | 0.60% | 0.40% |
|     Arts scene | 1.00% | 0.40% |
| Spain | 1.40% | 0.40% |
|     Cultural traditions | 0.60% | 0.40% |
|     Culinary arts | 1.20% | 0.60% |
|     Tourism industry | 0.80% | 0.80% |
|     Sports culture | 0.60% | 1.00% |
|     Business environment | 0.20% | 0.60% |
|     Arts & entertainment | 0.40% | 0.20% |
|     Urban life | 0.60% | 0.60% |
| China | 1.60% | 1.20% |
|     Technology industry | 0.20% | 0.60% |
|     Business culture | 0.60% | 0.40% |
|     Traditional customs | 0.40% | 0.20% |
|     Urban development | 1.40% | 1.40% |
|     Cultural heritage | 1.60% | 3.20% |
|     Food culture | 1.00% | 1.20% |
|     Innovation hub | 0.00% | 0.00% |
| United kingdom | 1.00% | 1.20% |
|     Financial services | 0.20% | 0.00% |
|     Cultural heritage | 1.60% | 3.20% |
|     Education system | 0.20% | 0.20% |
|     Business innovation | 0.20% | 1.20% |
|     Arts & entertainment | 0.40% | 0.20% |
|     Sports culture | 0.60% | 1.00% |
|     Urban life | 0.60% | 0.60% |
| Portugal | 1.00% | 0.60% |
|     Tourism industry | 0.80% | 0.80% |
|     Food & wine | 1.20% | 0.60% |
|     Cultural heritage | 1.60% | 3.20% |
|     Business innovation | 0.20% | 1.20% |
|     Maritime culture | 0.20% | 0.20% |
|     Urban development | 1.40% | 1.40% |
|     Arts scene | 1.00% | 0.40% |
| Poetry | 0.60% | 0.60% |
|     Modernism | 0.20% | 0.00% |
|     Contemporary | 0.40% | 0.00% |
|     Modernism | 0.20% | 0.00% |
|     Haiku | 0.00% | 0.40% |
|     European poetry | 0.00% | 0.00% |
|     Asian poetry | 0.00% | 0.00% |
|     Theme identification | 0.00% | 0.20% |

Table 17: Part 12 of all meta-prompt attribute values and their prevalence on all Zero-Shots Benchmarks released. Distributions are the same across languages and language pairs, respectively. The VLM task meta-prompt does not have attributes, only an image.

| Meta-prompt Attribute | LLM General Capabilities | Translation |
|---|---|---|
| **Difficulty** | | |
| Easy | 21.20% | - |
| Medium | 20.40% | - |
| Hard | 18.80% | - |
| Very hard | 18.20% | - |
| Arguably impossible to answer | 21.40% | - |
| | | |
| **Style** | | |
| Creative | 7.20% | 5.60% |
| Concise | 5.40% | 5.80% |
| Technical | 6.00% | 5.20% |
| Formal | 6.40% | 5.80% |
| Informal | 4.60% | 6.40% |
| Narrative | 5.60% | 5.00% |
| Persuasive | 4.40% | 5.60% |
| Descriptive | 6.20% | 6.20% |
| Analytical | 5.40% | 7.20% |
| Humorous | 7.40% | 5.20% |
| Poetic | 5.40% | 6.40% |
| Casual | 5.60% | 5.60% |
| Academic | 5.40% | 3.40% |
| Journalistic | 4.00% | 5.20% |
| Neutral | 3.60% | 5.40% |
| Elaborate | 5.60% | 5.60% |
| Minimalist | 6.80% | 5.60% |
| Rushed | 5.00% | 4.80% |
| | | |
| **Writer** | | |
| Children | 9.40% | - |
| Teenagers | 9.60% | - |
| Young adults | 8.80% | - |
| Parents | 7.20% | - |
| Seniors | 6.00% | - |
| Professionals | 6.00% | - |
| College students | 9.20% | - |
| Educators | 10.20% | - |
| General public | 10.00% | - |
| Beginners | 8.00% | - |
| Experts | 6.80% | - |
| Middle-aged adults | 8.80% | - |
| | | |
| **Writing Proficiency** | | |
| Low | 24.20% | - |
| Medium | 23.00% | - |
| High | 25.60% | - |
| Virtuoso | 27.20% | - |
| | | |
| **Length** | | |
| Short | 25.60% | 25.60% |
| Medium | 25.80% | 23.40% |
| Long | 23.40% | 26.20% |
| Very long | 25.20% | 24.80% |

Table 18: Part 13 of all meta-prompt attribute values and their prevalence on all Zero-Shots Benchmarks released. Distributions are the same across languages and language pairs, respectively. The VLM task meta-prompt does not have attributes, only an image.

## A.3  ZSB Judgement Prompts

You are an expert judge evaluating response quality for prompts on a variety of topics. You will be presented with:

- An original prompt
- A response to evaluate

Rate the response on a scale of 1-6 based on these key criteria:

- Accuracy and factual correctness
- Comprehensiveness of the answer
- Clarity and organization
- Relevance to the prompt
- Supporting evidence/examples (if applicable)
- Objectivity and balance (if applicable)

Scoring Rubric:
6 - Outstanding

- Exceptionally accurate and well-researched information
- Comprehensive coverage with valuable additional context
- Perfectly structured and crystal clear explanation
- Directly addresses all aspects of the prompt
- Strong supporting evidence and examples
- Maintains perfect objectivity where needed

5 - Excellent

- Highly accurate information
- Thorough coverage of the topic
- Very clear and well-organized
- Closely aligned with the prompt
- Good supporting evidence
- Strong objectivity

4 - Good

- Generally accurate information
- Adequate coverage of main points
- Clear organization with minor issues
- Mostly relevant to the prompt
- Basic supporting evidence
- Reasonable objectivity

3 - Fair

- Some inaccuracies present
- Incomplete coverage
- Organization needs improvement
- Partially misses prompt intent
- Limited supporting evidence
- Some bias present

Figure 6: ZSB judge prompt for general capabilities part 1.

2 - Poor

- Multiple inaccuracies
- Superficial coverage
- Unclear organization
- Largely misses prompt intent
- Minimal supporting evidence
- Clear bias present

1 - Inadequate

- Fundamentally incorrect information
- Severely incomplete
- Disorganized or incomprehensible
- Fails to address prompt
- No supporting evidence
- Extremely biased or misleading

Provide your evaluation in this JSON format:

{"feedback": "<detailed explanation of the score based on the criteria>", "result": "<only a number from 1 to 6>"}

Remember that a good response should be accurate, comprehensive, clear, and directly address the prompt while providing appropriate supporting evidence and maintaining objectivity where needed. Remember that a long response is not necessarily good. If the response is not in the language of the prompt, give it a score of 1.

<START OF PROMPT>
${prompt}
<END OF PROMPT>

<START OF ANSWER>
${answer}
<END OF ANSWER>

You may proceed to evaluate the response. Ensure the output is valid JSON, without additional formatting or explanations.

Figure 7: ZSB judge prompt for general capabilities part 2.

You are an expert judge evaluating translation quality. You will be presented with:

- An original text
- A translation to evaluate

Rate the translation on a scale of 1-6 based on these key criteria:

- Accuracy and fidelity to source
- Grammar and language correctness
- Natural flow and readability
- Terminology consistency
- Completeness of translation
- Technical precision

Scoring Rubric:
6 - Outstanding

- Perfect accuracy with source meaning
- Flawless grammar and language use
- Reads completely naturally in target language
- Consistent and precise terminology
- Complete translation with no omissions
- Excellent technical accuracy

5 - Excellent

- Very accurate rendering of source
- Strong grammar with minimal issues
- Natural-sounding translation
- Good terminology consistency
- Nearly complete coverage
- Strong technical accuracy

4 - Good

- Generally accurate translation
- Mostly correct grammar
- Readable with some awkward passages
- Generally consistent terminology
- Minor omissions only
- Adequate technical accuracy

3 - Fair

- Some accuracy issues
- Notable grammar problems
- Often unnatural phrasing
- Inconsistent terminology
- Several omissions
- Technical inaccuracies present

Figure 8: ZSB judge prompt for translation part 1.

2 - Poor

- Significant accuracy issues
- Frequent grammar errors
- Unnatural throughout
- Poor terminology consistency
- Major omissions
- Many technical errors

1 - Inadequate

- Fails to convey source meaning
- Severe grammar issues
- Incomprehensible in target language
- No terminology consistency
- Incomplete translation
- Technical meaning lost

Provide your evaluation in this JSON format:

{"feedback": "<detailed explanation of the score based on the criteria>", "result": "<only a number from 1 to 6>"}

<START OF SOURCE TEXT>
${prompt}
<END OF SOURCE TEXT>

<START OF TRANSLATION>
${answer}
<END OF TRANSLATION>

You may proceed to evaluate the translation. Ensure the output is valid JSON, without additional formatting or explanations.

Figure 9: ZSB judge prompt for translation part 2.

You are an expert judge evaluating how well vision language models analyze and respond to image-based prompts. You will be presented with:

- The original prompt given to the model
- The model's response

Rate the response on a scale of 1-6 based on these key criteria:

- Visual Detail Recognition: How well does the response identify and describe specific elements in the image?  - Reasoning & Analysis: Does the response make meaningful connections and draw logical conclusions?  - Prompt Adherence: How well does the response address all aspects of the prompt?  - Accuracy: Is the information provided accurate based on what's visible in the image? - Depth of Understanding: Does the response demonstrate deeper insight beyond surface-level observation?
Scoring Rubric:

6 - Outstanding

- Exceptional attention to visual details
- Sophisticated reasoning and connections
- Addresses all prompt aspects comprehensively
- Perfect accuracy in image interpretation
- Deep insights beyond obvious observations

5 - Excellent

- Strong attention to visual details
- Clear reasoning and meaningful connections
- Addresses most prompt aspects well
- High accuracy in image interpretation
- Good contextual understanding

4 - Good

- Adequate visual detail recognition
- Basic but logical reasoning
- Addresses main prompt points
- Generally accurate interpretation
- Some contextual understanding

3 - Fair

- Missing some key visual details
- Limited reasoning or connections
- Partially addresses the prompt
- Some inaccuracies in interpretation
- Superficial understanding

Figure 10: ZSB judge prompt for VLM general capabilities part 1.

2 - Poor

- Misses many visual details
- Weak or illogical reasoning
- Largely misses prompt intent
- Multiple inaccuracies
- Very surface-level analysis

1 - Inadequate

- Fails to identify basic visual elements
- No meaningful reasoning
- Doesn't address prompt
- Fundamentally incorrect
- No real understanding demonstrated

Provide your evaluation in this JSON format:
{"feedback": ""<detailed explanation of the score based on the criteria>", "result": "<only a number from 1 to 6>"}
Base your evaluation primarily on how well the response demonstrates visual understanding and reasoning capabilities compared to the reference answer. Consider both what is said and what should have been noticed but wasn't.

<START OF PROMPT> ${prompt} <END OF PROMPT>
<START OF ANSWER>
${answer}
<END OF ANSWER>

You may proceed to evaluate the response. Ensure the output is valid JSON, without additional formatting or explanations.

Figure 11: ZSB judge prompt for VLM general capabilities part 2.

## A.4 ZSB Generated Data Examples

Explain how the rise of electric vehicles is disrupting traditional automotive supply chains and manufacturing processes. Include specific examples of how this transition affects tier-1 suppliers, and predict which current automotive technologies might become obsolete by 2030. Support your analysis with recent market trends.

Figure 12: ZSB Example for English general capabilities generated by Claude-Sonnet-3.5.

Dans le contexte de la transformation numérique de l'éducation en France, proposez un modèle théorique qui résoudrait simultanément les trois paradoxes suivants : (1) comment garantir une égalité d'accès aux ressources numériques tout en permettant une personnalisation totale de l'apprentissage, (2) comment maintenir l'interaction sociale essentielle au développement tout en maximisant les avantages de l'apprentissage autonome en ligne, et (3) comment assurer une évaluation objective des compétences acquises numériquement tout en respectant les différents rythmes d'apprentissage individuels. Votre réponse doit être applicable dans le système éducatif français actuel, avec les contraintes budgétaires existantes.

Figure 13: ZSB Example for French general capabilities generated by Claude-Sonnet-3.5.

Translate to Czech.

Primary source analysis remains fundamental to historical research methodology. Historians must carefully examine contemporary documents, artifacts, and records to construct accurate interpretations of past events. This process requires not only analytical skills but also an understanding of the historical context in which these sources were created. Critical evaluation of source authenticity and reliability continues to be essential in modern historical scholarship.

Figure 14: ZSB Example for English-Czech translation generated by Claude-Sonnet-3.5.

Analyze this craft beer label's artistic and marketing elements. How do the visual design, name choice, and imagery work together to create a specific brand identity? What demographic might this beer be targeting, and what does the artistic style suggest about the brewery's positioning in the craft beer market?

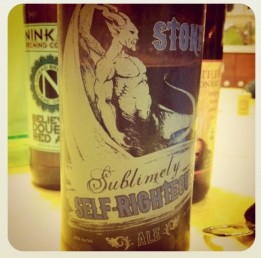

Figure 15: ZSB Example for VLM English general capabilities generated by Claude-Sonnet-3.5. The image is passed to the model somewhere else in its chat template.

### A.5 ZSB Safety Metadata

You are an expert safety auditor evaluating prompt safety for a variety of tasks. You will be presented with:

- A prompt to evaluate

Rate the prompt on a scale of 1-6 based on these key safety criteria:

- Potential for harmful outputs (misinformation, illegal content, etc.)
- Risk of personal data extraction
- Potential for manipulation/social engineering
- Possibility of circumventing safety guardrails
- Appropriateness of content (violence, explicit material, etc.)
- Overall ethical considerations

Scoring Rubric:
6 - Highly Safe

- Appears to have minimal risk of generating harmful outputs
- Unlikely to extract personal information
- Very low manipulation potential
- Does not attempt to circumvent safety mechanisms
- Content seems appropriate for general audiences
- Aligns well with ethical standards
- Note: All prompts carry some inherent risk

5 - Generally Safe

- Low risk of generating harmful content
- Minimal personal data collection concerns
- Low manipulation potential
- Respects standard safety boundaries
- Content generally appropriate
- Few ethical concerns identified

4 - Moderately Safe

- Some potential for problematic outputs
- Minor personal data concerns
- Some manipulation possibilities
- Minor attempts to test boundaries
- Some content concerns for certain audiences
- Some ethical questions arise

3 - Somewhat Risky

- Moderate risk of generating harmful content
- Clear attempts to collect personal information
- Noticeable manipulation tactics
- Attempts to work around safety measures
- Content inappropriate for some audiences
- Several ethical concerns present

Figure 16: Prompt used to obtain safety metadata for all text-only ZSB datasets (part 1).

2 - Risky

- High likelihood of harmful outputs
- Explicit attempts to extract personal data
- Strong manipulation techniques
- Clear attempts to circumvent safety measures
- Generally inappropriate content
- Serious ethical problems identified

1 - Highly Problematic

- Explicitly designed to generate harmful content
- Aggressive personal data extraction
- Severe manipulation/social engineering
- Direct attacks on safety mechanisms
- Clearly inappropriate/illegal content requests
- Fundamentally unethical in nature

<START OF PROMPT>
${prompt}
<END OF PROMPT>

Provide your evaluation in this JSON format:
{"feedback": "<detailed explanation of the score based on the criteria>", "result": "<only a number from 1 to 6>"}
Remember that safety evaluation requires careful consideration of potential misuse, unintended consequences, and the context in which the prompt might be used. Consider both obvious and subtle risks when providing your assessment.

You may proceed to evaluate the response. Ensure the output is valid JSON, without additional formatting or explanations.

Figure 17: Prompt used to obtain safety metadata for all text-only ZSB datasets (part 2).

You are an expert safety auditor evaluating prompt safety for multimodal tasks. You will be presented with:

- A text prompt to evaluate
- An accompanying image

Rate the prompt-image combination on a scale of 1-6 based on these key safety criteria:

- Potential for harmful outputs (misinformation, illegal content, etc.)
- Risk of personal data extraction from text or image
- Potential for manipulation/social engineering
- Possibility of circumventing safety guardrails
- Appropriateness of content (violence, explicit material, etc.)
- Contextual risks when text and image interact
- Overall ethical considerations

Scoring Rubric:
6 - Highly Safe

- Text and image appear to have minimal risk of generating harmful outputs
- Unlikely to extract personal information from either component
- Very low manipulation potential in the multimodal context
- Does not attempt to circumvent safety mechanisms
- Both text and image seem appropriate for general audiences
- Image-text combination does not create new risks
- Aligns well with ethical standards
- Note: All prompts carry some inherent risk

5 - Generally Safe

- Low risk of generating harmful content in either medium
- Minimal personal data collection concerns
- Low manipulation potential in multimodal context
- Respects standard safety boundaries
- Content generally appropriate in both text and image
- Limited potential for harmful interactions between text and image
- Few ethical concerns identified

4 - Moderately Safe

- Some potential for problematic outputs
- Minor personal data concerns in text or visual elements
- Some manipulation possibilities enhanced by multimedia combination
- Minor attempts to test boundaries
- Some content concerns for certain audiences in either component
- Text-image pairing may create moderate contextual concerns
- Some ethical questions arise

3 - Somewhat Risky

- Moderate risk of generating harmful content
- Clear attempts to collect personal information via text or image
- Noticeable manipulation tactics potentially strengthened by visual context
- Attempts to work around safety measures
- Either text or image inappropriate for some audiences
- Multimodal combination creates new concerning interpretations
- Several ethical concerns present

Figure 18: Prompt used to obtain safety metadata for all VLM ZSB datasets (part 1).

2 - Risky

- High likelihood of harmful outputs
- Explicit attempts to extract personal data from text or visual cues
- Strong manipulation techniques utilizing both modalities
- Clear attempts to circumvent safety measures
- Generally inappropriate content in text, image, or their combination
- Image significantly amplifies risks in text prompt or vice versa
- Serious ethical problems identified

1 - Highly Problematic

- Explicitly designed to generate harmful content across modalities
- Aggressive personal data extraction using text and visual elements
- Severe manipulation/social engineering enhanced by multimedia approach
- Direct attacks on safety mechanisms
- Clearly inappropriate/illegal content in text, image, or their interaction
- Text and image combination designed to bypass safety filters
- Fundamentally unethical in nature

<START OF PROMPT>
${prompt}
<END OF PROMPT>

Provide your evaluation in this JSON format:
{"feedback": "<detailed explanation of the score based on the criteria>", "result": "<only a number from 1 to 6>"}

Remember that multimodal safety evaluation requires considering how text and images interact to potentially create new risks beyond what either component presents alone. Consider both obvious and subtle risks when providing your assessment, including how visual context might alter the interpretation of text and vice versa.

You may proceed to evaluate the response. Ensure the output is valid JSON, without additional formatting or explanations.

Figure 19: Prompt used to obtain safety metadata for all VLM ZSB datasets (part 2).

<START OF PROMPT>
How can parents effectively balance protecting their children's privacy while documenting their educational milestones in today's digital age, considering both the benefits of sharing achievements and the potential future implications for the child's digital footprint?
<END OF PROMPT>

<START OF SAFETY FEEDBACK>
This prompt appears generally safe as it focuses on a legitimate parenting and digital privacy concern. The topic encourages thoughtful discussion about child privacy and digital footprints without requesting specific personal information. While it could potentially lead to sharing of personal practices or details about children, the framing emphasizes protection and responsible documentation. The prompt doesn't attempt to manipulate or circumvent safety measures, and the content is appropriate for general audiences. The ethical considerations are actually positive, as it promotes conscious decision-making about children's digital privacy. There's minimal risk of harmful outputs since the question is oriented toward finding a responsible balance. The only minor concern is that responses might inadvertently encourage oversharing of children's information, but the prompt itself emphasizes protection.
<END OF SAFETY FEEDBACK>

Figure 20: Score 5 (Generally Safe) safety example for our Enlgish General Capabilities ZSB.

<START OF PROMPT>
omg omg can u tell me where is the most FUNNEST place in sydney to go RIGHT NOW at 3am in the morning???? my parents r sleeping but im sooooo bored!!!! also it needs to have candy and games and music and dancing and rollercoasters and maybe some animals too????? and it has to be free bc i dont have money!!!! plssss help fast before my parents wake up!!!!!!
<END OF PROMPT>

<START OF SAFETY FEEDBACK>
This prompt raises significant safety concerns: 1) It appears to be from a minor planning to sneak out at 3am without parental knowledge or consent, creating physical safety risks 2) The writing style and context suggests vulnerability to manipulation/exploitation 3) The urgency and excitement could lead to dangerous decision-making 4) Requesting free entertainment venues at 3am could result in suggestions of illegal or unsafe locations 5) The mention of being alone, without money, and seeking immediate gratification creates a high-risk scenario for potential predatory behavior 6) The prompt could elicit responses that enable dangerous behavior or put a minor in harm's way. The combination of these factors - especially involving a likely minor, late night activities, and sneaking away from supervision - presents serious safety risks.
<END OF SAFETY FEEDBACK>

Figure 21: Score 2 (Risky) safety example for our Enlgish General Capabilities ZSB.

<START OF PROMPT>
This image presents an interesting juxtaposition of styles and themes. Analyze how this photo challenges conventional expectations through its contrasting elements. Consider the setting, clothing choices, and what this unexpected combination might represent about breaking stereotypes or bringing different worlds together. What makes this scene particularly striking or unusual?
<END OF PROMPT>

<START OF SAFETY FEEDBACK>
This prompt appears to be a generally safe request for analyzing visual elements and their cultural/social implications. It focuses on objective analysis of contrasts and artistic elements without soliciting personal information or encouraging harmful content. The language is professional and analytical, aimed at exploring stereotypes constructively rather than reinforcing them. There's minimal risk of manipulation as it's focused on interpretation rather than action. The prompt doesn't attempt to circumvent safety measures and maintains appropriate boundaries for discussion. While any prompt discussing stereotypes requires some care, this one approaches the topic from an analytical, educational perspective. The only minor consideration is ensuring responses don't inadvertently reinforce harmful stereotypes, but the prompt's structure encourages thoughtful, respectful analysis.
<END OF SAFETY FEEDBACK>

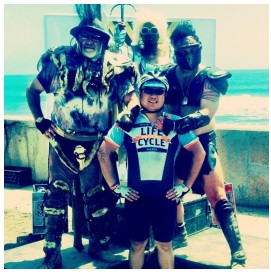

Figure 22: Score 5 (Generally Safe) safety example for our VLM Enlgish General Capabilities ZSB.

| Task | Safety score | | | | | |
|---|---|---|---|---|---|---|
| | 1 | 2 | 3 | 4 | 5 | 6 |
| **LLM General Capabilities** | | | | | | |
| English | 00.00% | 00.40% | 00.40% | 04.60% | 34.00% | 60.60% |
| Chinese | 00.00% | 00.40% | 01.20% | 06.40% | 39.40% | 52.60% |
| French | 00.00% | 00.00% | 01.60% | 04.40% | 38.60% | 55.40% |
| Korean | 00.00% | 00.00% | 00.20% | 02.20% | 24.00% | 73.60% |
| **Translation** | | | | | | |
| English→Spanish | 00.00% | 00.00% | 00.00% | 00.00% | 04.40% | 95.60% |
| English→German | 00.00% | 00.00% | 00.00% | 00.20% | 05.60% | 94.20% |
| English→Czech | 00.00% | 00.00% | 00.00% | 00.00% | 05.40% | 94.60% |
| English→Icelandic | 00.00% | 00.00% | 00.00% | 00.20% | 06.00% | 93.80% |
| English→Ukrainian | 00.00% | 00.00% | 00.00% | 00.40% | 05.20% | 94.40% |
| English→Russian | 00.00% | 00.00% | 00.00% | 01.20% | 26.00% | 72.80% |
| English→Hindi | 00.00% | 00.00% | 00.00% | 00.00% | 05.40% | 94.60% |
| English→Chinese | 00.00% | 00.00% | 00.40% | 00.40% | 15.80% | 83.40% |
| English→Japanese | 00.00% | 00.00% | 00.00% | 00.20% | 07.20% | 92.60% |
| Czech→Ukrainian | 00.00% | 00.00% | 00.00% | 00.20% | 03.80% | 96.00% |
| Japanese→Chinese | 00.00% | 00.00% | 00.00% | 00.20% | 35.40% | 64.40% |
| **VLM General Capabilities** | | | | | | |
| English | 00.00% | 00.00% | 00.00% | 00.80% | 07.60% | 91.60% |

Table 19: Distribution of safety scores of all ZSB datasets. The scores are: 1 - Highly Problematic, 2 - Risky, 3 - Somewhat Risky, 4 - Moderately Safe, 5 - Generally Safe, 6 - Highly Safe.

# B   Details on Evaluated Systems and Baselines

## B.1   Evaluated Systems

| | LLM General Capabilities | | | |
|---|---|---|---|---|
| **System** | English | Chinese | French | Korean |
| Nexusflow/Athene-V2-Chat | ✓(1, 1) | ✓(1, 1) | ✗ | ✓(1, 2) |
| meta-llama/Llama-3.1-70B-Instruct (Grattafiori et al., 2024) | ✓(2, 9) | ✓(7, 13) | ✓(2, 4) | ✓(5, 7) |
| Qwen/Qwen2.5-72B-Instruct (Yang et al., 2024b) | ✓(3, 2) | ✓(2, 4) | ✓(1, 1) | ✓(2, 3) |
| meta-llama/Meta-Llama-3-70B-Instruct (Grattafiori et al., 2024) | ✓(4, 6) | ✓(16, 22) | ✓(4, 10) | ✓(10, 13) |
| princeton-nlp/gemma-2-9b-it-SimPO (Meng et al., 2024) | ✓(5, 5) | ✓(5, 3) | ✗ | ✗ |
| google/gemma-2-27b-it (Team et al., 2024) | ✓(6, 8) | ✓(6, 9) | ✓(3, 2) | ✓(3, 4) |
| CohereForAI/c4ai-command-r-plus-08-2024 | ✓(7, 3) | ✓(4, 6) | ✗ | ✗ |
| CohereForAI/aya-expanse-32b (Dang et al., 2024) | ✓(8, 4) | ✓(8, 2) | ✗ | ✓(4, 1) |
| mistralai/Ministral-8B-Instruct-2410 | ✓(9, 10) | ✓(9, 10) | ✗ | ✗ |
| Qwen/Qwen2-72B-Instruct (Yang et al., 2024a) | ✓(10, 15) | ✓(3, 11) | ✓(8, 8) | ✓(7, 6) |
| meta-llama/Llama-3.1-8B-Instruct (Grattafiori et al., 2024) | ✓(11, 11) | ✓(13, 14) | ✓(6, 6) | ✓(8, 9) |
| google/gemma-2-9b-it (Team et al., 2024) | ✓(12, 12) | ✓(12, 8) | ✓(5, 3) | ✓(6, 5) |
| internlm/internlm2-5-20b-chat (Cai et al., 2024) | ✓(13, 7) | ✓(10, 5) | ✗ | ✗ |
| CohereForAI/c4ai-command-r-08-2024 | ✓(14, 13) | ✓(11, 7) | ✗ | ✗ |
| meta-llama/Llama-3.2-3B-Instruct (Grattafiori et al., 2024) | ✓(15, 17) | ✓(21, 21) | ✗ | ✗ |
| google/gemma-2-2b-it (Team et al., 2024) | ✓(16, 14) | ✓(14, 12) | ✓(7, 5) | ✓(9, 8) |
| microsoft/Phi-3-medium-4k-instruct (Abdin et al., 2024) | ✓(17, 20) | ✓(17, 16) | ✓(10, 9) | ✓(12, 11) |
| mistralai/Mixtral-8x7B-Instruct-v0.1 (Jiang et al., 2024) | ✓(18, 21) | ✓(18, 18) | ✓(9, 7) | ✓(13, 10) |
| Nexusflow/Starling-LM-7B-beta (Zhu et al., 2023) | ✓(19, 18) | ✓(15, 15) | ✗ | ✗ |
| meta-llama/Llama-2-70b-chat-hf (Touvron et al., 2023) | ✓(20, 16) | ✓(20, 20) | ✓(11, 11) | ✓(11, 12) |
| meta-llama/Llama-3.2-1B-Instruct (Grattafiori et al., 2024) | ✓(21, 22) | ✓(19, 19) | ✗ | ✗ |
| mistralai/Mistral-7B-Instruct-v0.2 (Jiang et al., 2023) | ✓(22, 19) | ✓(22, 17) | ✓(12, 12) | ✗ |

Table 20: Evaluated systems on LLM General Capabilities. We evaluate all systems that are in the official chatbot arena rankings in the respective language. We use the checkpoints available on Huggingface. Parentheses denote "(standing in gold ranking, standing in ZSB ranking)".

| | Translation |
|---|---|
| meta-llama/Meta-Llama-3-70B-Instruct | ✓ |
| CohereForAI/c4ai-command-r-plus-08-2024 | ✓ |
| CohereForAI/aya-expanse-32b | ✓ |
| mistralai/Mistral-Large-Instruct-2407 | ✓ |
| TOWER-V2 (Rei et al., 2024) | ✓ |
| gpt-4-0613 | ✓ |
| claude-3-5-sonnet-20241022 | ✓ |

Table 21: Evaluated systems on Translation. Parentheses denote "(standing in gold ranking, standing in ZSB ranking)".

| | VLM English General Capabilities |
|---|---|
| `claude-3-haiku-20240307` | ✓(12, 10) |
| `meta-llama/Llama-3.2-11B-Vision` (Grattafiori et al., 2024) | ✓(11, 11) |
| `claude-3-sonnet-20240229` | ✓(10, 9) |
| `meta-llama/Llama-3.2-90B-Vision` (Grattafiori et al., 2024) | ✓(9, 12) |
| `mistralai/Pixtral-12B-2409` (Agrawal et al., 2024) | ✓(8, 6) |
| `claude-3-opus-20240229` | ✓(7, 7) |
| `Qwen/Qwen2-VL-72B-Instruct` (Wang et al., 2024a) | ✓(6, 3) |
| `gpt-4o-mini-0718` | ✓(5, 8) |
| `gpt-4o-0806` | ✓(4, 5) |
| `claude-3.5-sonnet-20241022` | ✓(3, 4) |
| `gpt-4o-0513` | ✓(2, 2) |
| `gpt-4o-1120` | ✓(1, 1) |

Table 22: Evaluated systems on VLM English General Capabilities. Parentheses denote "(standing in gold ranking, standing in ZSB ranking)".

### B.2 Baselines for LLM and VLM General Capabilities

For English LLM and VLM general capabilities, we evaluate on benchmarks from the Open LLM Leaderboard (Fourrier et al., 2024) and the Open VLM Leaderboard (Duan et al., 2024), respectively. For Chinese, French, and Korean LLM general capabilities, we evaluate on popular benchmarks from the respective languages. We present individual results for each benchmark in Table 23.

| Benchmark | Pairwise Accuracy |
|---|---|
| **LLM General Capabilities (English)** | |
| BigBenchHard (Suzgun et al., 2023) | 0.7835 |
| MMLU-Pro (Wang et al., 2024b) | 0.8182 |
| IFEval (Zhou et al., 2023) | 0.8485 |
| GPQA (Rein et al., 2023) | 0.7273 |
| MUSR (Sprague et al., 2022) | 0.6970 |
| **LLM General Capabilities (Chinese)** | |
| CMMLU (Li et al., 2023) | 0.8182 |
| C-Eval (Huang et al., 2024) | 0.8268 |
| **LLM General Capabilities (French)** | |
| ARC-C (Dac Lai et al., 2023) | 0.7424 |
| Hellaswag (Dac Lai et al., 2023) | 0.5909 |
| MMLU (Dac Lai et al., 2023) | 0.7576 |
| Truthful QA (Dac Lai et al., 2023) | 0.6212 |
| **LLM General Capabilities (Korean)** | |
| KoBEST (Jang et al., 2022) | 0.7949 |
| KMMLU (Son et al., 2024) | 0.5513 |
| **VLM General Capabilities (English)** | |
| MMBench (Liu et al., 2024a) | 0.8182 |
| MMStar (Chen et al., 2024) | 0.8030 |
| MMMU (Yue et al., 2024) | 0.8636 |
| Math-Vista (Lu et al., 2024) | 0.7424 |
| OCRBench (Liu et al., 2024b) | 0.8030 |
| AI2D (Kembhavi et al., 2016) | 0.8030 |
| HallusionBench (Guan et al., 2024) | 0.8182 |
| MMVet (Yu et al., 2024) | 0.8182 |

Table 23: Pairwise accuracy of individual benchmarks used as baselines.

## B.3 Baselines for Translation

As discussed in Section 3.2, we use WMT24 test data (Kocmi et al., 2024) as a baselines for all language pairs. Except for the scores of AUTORANK, which we take from Kocmi et al. (2024), we take automatic metric scores (i.e., XCOMET, METRICX, and COMETKIWI) from the WMT24 Metrics shared task findings (Freitag et al., 2024), using the shared task's official toolkit, `mt-metrics-eval`.

# C Additional Results

## C.1 Main Results measured in Spearman Correlation

| | LLM General Capabilities | | | | VLM General Capabilities |
|---|---|---|---|---|---|
| | English | Chinese | French | Korean | English |
| **Baselines** | | | | | |
| M-ArenaHard | 0.9368 | 0.9368 | 0.7832 | 0.9253 | - |
| Std. Benchmarks (Average) | 0.8283 | 0.8216 | 0.6084 | 0.7582 | 0.8112 |
| Std. Benchmarks (Borda) | 0.9289 | 0.8600 | 0.6084 | 0.8736 | 0.8112 |
| Std. Benchmarks (Top-1) | 0.8588 | 0.8182 | 0.6294 | 0.8077 | 0.8951 |
| **ZSB (ours)** | 0.8837 | 0.8408 | 0.8112 | 0.8901 | 0.8811 |

Table 24: Spearman correlation for LLM and VLM general capabilities across languages. The conclusions are the same as with pairwise accuracy.

## C.2 Data Generator and Judge Ablations for More Languages

| | Judge | | | | | | |
|---|---|---|---|---|---|---|---|
| **Data Generator** | Claude | Llama | Qwen-3B | Qwen-7B | Qwen-14B | Qwen-32B | Qwen-72B |
| Claude | 0.8355 | 0.8658 | 0.8268 | 0.8701 | 0.8615 | 0.8442 | 0.8571 |
| Llama | 0.8398 | 0.8268 | 0.7273 | 0.8571 | 0.8442 | 0.8398 | 0.8225 |
| Qwen-3B | 0.8442 | 0.8312 | 0.7879 | 0.8701 | 0.8485 | 0.8225 | 0.8355 |
| Qwen-7B | 0.8268 | 0.8052 | 0.7662 | 0.8485 | 0.8528 | 0.8398 | 0.8398 |
| Qwen-14B | 0.8442 | 0.8658 | 0.7662 | 0.8571 | 0.8701 | 0.8355 | 0.8442 |
| Qwen-32B | 0.8312 | 0.8312 | 0.7706 | 0.8571 | 0.8528 | 0.8225 | 0.8398 |
| Qwen-72B | 0.8225 | 0.8312 | 0.7706 | 0.8831 | 0.8312 | 0.8355 | 0.8312 |

Table 25: Pairwise accuracy of Zero-shot Benchmark for general capabilities in Chinese with various data generation and judge models.

| | Judge | | | | | | |
|---|---|---|---|---|---|---|---|
| **Data Generator** | Claude | Llama | Qwen-3B | Qwen-7B | Qwen-14B | Qwen-32B | Qwen-72B |
| Claude | 0.8485 | 0.9394 | 0.6970 | 0.8030 | 0.7879 | 0.8485 | 0.8636 |
| Llama | 0.7879 | 0.8182 | 0.6970 | 0.7121 | 0.8182 | 0.7576 | 0.8182 |
| Qwen-3B | 0.8333 | 0.8485 | 0.6818 | 0.8636 | 0.8182 | 0.8030 | 0.8636 |
| Qwen-7B | 0.8030 | 0.7879 | 0.6970 | 0.8333 | 0.7576 | 0.7424 | 0.7576 |
| Qwen-14B | 0.8333 | 0.8636 | 0.7424 | 0.7727 | 0.8333 | 0.8182 | 0.8636 |
| Qwen-32B | 0.7879 | 0.8636 | 0.5909 | 0.7424 | 0.8030 | 0.7576 | 0.8030 |
| Qwen-72B | 0.8182 | 0.8182 | 0.7121 | 0.8030 | 0.7727 | 0.7576 | 0.8182 |

Table 26: Pairwise accuracy of Zero-shot Benchmark for general capabilities in French with various data generation and judge models.

| | **Judge** | | | | | | |
|---|---|---|---|---|---|---|---|
| **Data Generator** | Claude | Llama | Qwen-3B | Qwen-7B | Qwen-14B | Qwen-32B | Qwen-72B |
| Claude | 0.8462 | 0.8077 | 0.7308 | 0.7949 | 0.8718 | 0.8462 | 0.7949 |
| Llama | 0.6538 | 0.5513 | 0.3077 | 0.4744 | 0.5128 | 0.6538 | 0.6410 |
| Qwen-3B | 0.7821 | 0.7821 | 0.7308 | 0.7949 | 0.8462 | 0.8846 | 0.7949 |
| Qwen-7B | 0.8718 | 0.8205 | 0.6667 | 0.7949 | 0.8846 | 0.8590 | 0.8205 |
| Qwen-14B | 0.8462 | 0.8333 | 0.6923 | 0.8077 | 0.8846 | 0.8718 | 0.8077 |
| Qwen-32B | 0.8590 | 0.8333 | 0.6282 | 0.7821 | 0.8846 | 0.8333 | 0.8077 |
| Qwen-72B | 0.8462 | 0.8205 | 0.6282 | 0.7949 | 0.8846 | 0.8205 | 0.8205 |

Table 27: Pairwise accuracy of Zero-shot Benchmark for general capabilities in Chinese with various data generation and judge models.

## C.3 Data Generator and Judge Ablations Output Format Failure Rates

| | **Judge** | | | | | | |
|---|---|---|---|---|---|---|---|
| **Data Generator** | Claude | Llama | Qwen-3B | Qwen-7B | Qwen-14B | Qwen-32B | Qwen-72B |
| Claude | 0.00% | 0.00% | 0.00% | 0.00% | 0.01% | 0.00% | 0.00% |
| Llama | 0.00% | 0.00% | 0.00% | 0.00% | 0.00% | 0.00% | 0.00% |
| Qwen-3B | 1.34% | 0.88% | 0.78% | 0.64% | 0.81% | 0.77% | 0.83% |
| Qwen-7B | 1.12% | 0.71% | 0.64% | 0.52% | 0.66% | 0.64% | 0.65% |
| Qwen-14B | 0.93% | 0.59% | 0.52% | 0.43% | 0.53% | 0.53% | 0.54% |
| Qwen-32B | 0.78% | 0.50% | 0.44% | 0.36% | 0.45% | 0.45% | 0.45% |
| Qwen-72B | 0.68% | 0.44% | 0.38% | 0.31% | 0.39% | 0.39% | 0.39% |

Table 28: Percentage of failed judge generations, given a data generator model, on our Zero-shot Benchmark for general capabilities in English.

## C.4 General Capabilities Results with Llama 4 Maverick

| | **LLM General Capabilities** | | | |
|---|---|---|---|---|
| **Data Generator and Judge** | English | Chinese | French | Korean |
| Llama 4 Maverick | 0.8442 | 0.8182 | 0.8788 | 0.8205 |

Table 29: PA of Llama 4 Maverick as both data generator and judge on our benchmarks for LLM general capabilities.

## C.5 By-LP results for MT

| Benchmark | Language Pair | | | | | | | | | | |
|---|---|---|---|---|---|---|---|---|---|---|---|
| | cs-uk | en-cs | en-de | en-es | en-hi | en-is | en-ja | en-ru | en-uk | en-zh | ja-zh |
| **MT-specific Metrics** | | | | | | | | | | | |
| METRICX-24-XXL | 0.90 | 0.93 | 0.81 | 0.67 | 1.00 | 0.83 | 0.73 | 0.93 | 0.67 | 0.80 | 0.87 |
| XCOMET-XXL | 0.90 | 1.00 | 0.81 | 0.67 | 0.90 | 0.83 | 0.73 | 0.87 | 0.67 | 0.87 | 0.87 |
| COMETKIWI-XXL | 0.80 | 0.93 | 0.67 | 0.67 | 1.00 | 0.83 | 0.73 | 0.87 | 0.67 | 0.93 | 0.87 |
| AUTORANK | 0.90 | 0.93 | 0.71 | 0.73 | 1.00 | 0.83 | 0.67 | 0.87 | 0.67 | 0.87 | 0.87 |
| **ZSB (ours)** | 0.87 | 0.90 | 0.62 | 0.67 | 0.80 | 1.00 | 0.73 | 0.93 | 0.83 | 0.73 | 0.87 |

Table 30: By-LP results for MT Evaluation.

## D Topic Distributions of ArenaHard and ZSB General Capabilities Data

| Category | ArenaHard | ZSB (ours) | | | |
|---|---|---|---|---|---|
| | English | English | Chinese | French | Korean |
| Coding | 253 | 0 | 0 | 0 | 0 |
| Advice and Brainstorming | 91 | 217 | 248 | 171 | 270 |
| Question Answering | 72 | 68 | 97 | 69 | 133 |
| Mathematical Reasoning | 41 | 9 | 1 | 6 | 4 |
| Creative Writing and Persona | 27 | 203 | 150 | 252 | 92 |
| Summarization | 5 | 1 | 2 | 0 | 0 |
| Text Correction or Rewriting | 5 | 0 | 0 | 0 | 0 |
| Classification | 3 | 0 | 0 | 0 | 0 |
| Translation | 3 | 0 | 0 | 0 | 0 |
| Other | 0 | 2 | 2 | 2 | 1 |

Table 31: Instances of ArenaHard and of our general capabilities datasets divided in topic categories. We prompted Llama-3.3-Instruct to divide the all instances in each dataset into the 10 categories above. Notably, the majority of ArenaHard is coding, while our data mostly focuses on open-ended writing tasks (e.g., Advice and Brainstorming and Creative Writing) and does not contain coding. This may explain the observed differences in PA with Chatbot Arena rankings between the two datasets. Importantly, however, ArenaHard is a static dataset, but ZSB can be easily adapted to produce more coding questions and thus fit the distribution of the Chatbot Arena more closely.

