# OpenReview forum: "Zero-shot Benchmarking: A Framework for Flexible and Scalable Automatic Evaluation of Language Models"
_colmweb.org/COLM/2025/Conference — COLM 2025_

### Official Review · Reviewer_PEQ1 · 2025-04-15

**Rating:** 7
**Confidence:** 3
**Ethics Flag:** 1

**Summary:**

This paper proposes a zero-shot evaluation framework (ZSB), where Claude3.5-sonnet is used to create a test set, several other LLMs are tested, and  Claude is again used as a judge LLM to evaluate their performance. The paper concentrates on three tasks: open-ended prompt answering (English, Chinese, French, Korean), translation (11 language pairs), and multi-modal prompt answering (English).

After creating the data, the authors prompt a number of LLMs to respond to the tasks, and then use the judge LLM to give direct assessments for each response. The systems are ranked according to their average performance and this ranking is compared to a separate ranking based on human judgments of common benchmark datasets. The Pairwise Accuracy correlation suggests that ZSB correlates strongly with human judgments. The authors proceed to perform a detailed ablation study.

**Questions To Authors:**

- Does the inclusion of the meta-prompt attributes actually lead to the creation of corresponding tasks examples? It seems that this aspect has not been evaluated.

- Do the generated reference answers correctly complete the tasks?

- What are the tasks generated and what are their basic statistics?
    - distribution of task types - short-form question answering,
    - length of tasks and their generated references
    - the percentage of references that correctly complete the task
    - the percentage of meta-prompt attributes that are correctly modeled in the task

- If I've understood correctly, you compare system rankings created by judge LLMs on your generated data to system rankings created via human/metric judgments on other datasets. Is this really a useful comparison? Shouldn't you instead compare system rankings created via human judgments on your generated data to perform a true metaevaluation?


- The paper your cite for Pairwise Accuracy (Thompson et al., 2024) criticizes many aspects of Pairwise Accuracy and proposes a better solution: Soft Pairwise Accuracy. Why don't you use this measure instead of PA?

**Reasons To Accept:**

- The paper is well written and generally easy to follow.
- The authors include a large amount of meta-prompt attributes to create varied tasks in a controlled way.
- The paper contains a detailed analysis of many important variables: using open models to generate/judge, direct vs pairwise assessment, dataset size, or variety of meta-prompt attributes.

**Reasons To Reject:**

- The generated data are not analyzed or explained in any depth. The paper does not tell the reader what kind of prompts were created, how well the generated references complete these tasks, or provide any statistic other than the number of examples generated per task (500). As such, aside from translation abilities, it is unclear what abilities this methodology really evaluates.

- The methodology has several limiting requirements: 1) strong instruction-following model capable of generating fluent text, 2) a strong judge LLM capable of giving meaningful direct assessments, 3) tasks that can be assessed meaningfully via direct assessment. These requirements mean that for many languages and tasks, this methodology is likely to perform poorer than reported here. It would strengthen the paper to include experiments on an under-resourced language or structured prediction task to determine if this methodology is able to deal with adverse conditions.

---

> ### Author Response · Authors · 2025-06-01
> **Response (2/2)**
>
> >Does the inclusion of the meta-prompt attributes actually lead to the creation of corresponding tasks examples? It seems that this aspect has not been evaluated. [...] the percentage of meta-prompt attributes that are correctly modeled in the task
>
> This is a great question! We prompted Claude about whether each instance of all our LLM general capabilities benchmarks (across all languages) abided by the attributes passed in the meta prompt (i.e., binary classification). The results are the following:
>
> | | English | Chinese | French | Korean |
> |---|---|---|---|---|
> | Instance % agreement with meta-prompt attributes | 95.8 | 97.4 | 99.6 | 99.0 |
>
> The table shows the vast majority of generated instances model the meta-prompt attributes correctly on all languages.
>
> We are happy to include this analysis for the rest of our benchmarks in the final version (as well as to release the prompts and the code used). Thank you for raising this.
>
> >Do the generated reference answers correctly complete the tasks? [...] the percentage of references that correctly complete the task
> Thank you for raising this. We ran Qwen2.5-72B-Instruct as a judge on top of Claude’s references for all our LLM General Capabilities benchmarks. In the following table, we show the percentage of times the reference receives the highest and the second highest scores:
>
> || English | Chinese | French | Korean |
> |---|---|---|---|---|
> |% of perfect references|7.4|26.8|5.2|2.0|
> |% of score 5|89.2|68.6|86.6|88.2|
>
> While the vast majority of references are high-quality, few are perfect, according to this judge. This perhaps explains why the gains in PA of including a reference during evaluation are not very high (see Figure 2 b)). This is an interesting finding, which we will include in the final version. Thank you for raising this.
>
> >If I've understood correctly, you compare system rankings created by judge LLMs on your generated data to system rankings created via human/metric judgments on other datasets. Is this really a useful comparison? Shouldn't you instead compare system rankings created via human judgments on your generated data to perform a true metaevaluation?
>
> Comparing system rankings created via human judgments on our generated data with our predicted system rankings would be evaluating **solely** the evaluation component of ZSB, rather than the entire framework. Note that ZSB yields an entire benchmark (i.e., **test data** + **an evaluation method**), so, as such, we must compare it directly with other benchmarks as a whole. Your suggestion is indeed valuable in the context of evaluating the **evaluation method**, but it would not take into consideration the **test data** generation component of ZSB.
>
> Similarly to other prominent efforts in this area (e.g., Arena-Hard-Auto [1]), we pose that the most appropriate way of doing this evaluation is thus to choose a reliable gold ranking for a given task/set of capabilities (we describe our choices in Section 3.2), and to compare those rankings against system rankings obtained by different benchmarks. For example, M-Arena-Hard sources the test data from the Chatbot Arena and yields rankings according to a LLM-as-a-Judge evaluation via pairwise comparison (the final model scores are obtained via the Bradley & Terry model); ZSB, on the other hand, sources the test data from LLM-generated answers to a human metaprompt and yields rankings according to a LLM-as-a-Judge evaluation (we found a direct assessment rubric to be the most effective). As such, when we compare ZSB to Arena-Hard-Auto, we are indeed comparing the full “pipeline”, including test data sourcing and evaluation method. We hope this clarification answers your questions.
>
> [1] Li, T., Chiang, W. L., Frick, E., Dunlap, L., Wu, T., Zhu, B., ... & Stoica, I. (2024). From Crowdsourced Data to High-Quality Benchmarks: Arena-Hard and BenchBuilder Pipeline. CoRR.
>
>
> >The paper you cite for Pairwise Accuracy (Thompson et al., 2024) criticizes many aspects of Pairwise Accuracy and proposes a better solution: Soft Pairwise Accuracy. Why don't you use this measure instead of PA?
>
> Thank you for pointing this out. Soft PA improves upon PA by evaluating metrics not only on whether they rank systems accurately, but also on whether they do it with the same confidence as that suggested by the gold rankings. However, in order to do this, soft PA requires instance-level scores. These scores must be obtained from the same instances, otherwise they are not comparable. This is the case in WMT, where Thompson et al. (2024) propose the metric, since gold and predicted rankings are obtained from the same data, just different metrics (humans for the gold rankings, and automatic metrics for predicted rankings). With ZSB, we create our own instances, so the metric we use must be instance-agnostic, justifying choosing PA over soft PA. We hope this clarifies this matter.

---

> > ### Comment · Reviewer_PEQ1 · 2025-06-05
> >
> > Thank you for the response.
> >
> > Regarding the lack of information on the generated data, after your response, it is still not 100% clear how many test instances were used. In the paper you mention that for each task (you mention 6 tasks), you create 500 test instances, which should be 3000 test instances. However, this information isn't mentioned in the paper or the response. It would be helpful to clear this up. The metaprompts give an idea of what you want to be generated, not what was truly generated or whether the generated answer is correct.
> >
> > The analysis from your response shows that the general capabilities set of your benchmark is heavily biased towards open-ended advice and creative writing, while Arena-Hard is biased towards coding. This seems to suggest that your benchmarks are likely evaluating different model capabilities, and perhaps not really comparable. That is, if a model ranks better on Arena-Hard, it might be that it is better at coding, while a high-ranking model on your benchmark might be better at creative-writing and brainstorming. I think this is another import reason to detail the kinds of tasks that are created in your benchmark.
> >
> > Regarding the complexity analysis, I have doubts about the validity of using Llama-3.3-70B to determine the complexity of the tasks in both benchmarks.
> >
> > The translation experiments on low-resource languages would be interesting. However, given that one of the main advantages of your method is that it is easy to implement in a new language, running the general LLM capabilities experiment on a low-resource language would strengthen your claims without introducing a large burden.

---

> > > ### Author Response · Authors · 2025-06-05
> > > **Response**
> > >
> > > Thank you for your response and for the additional comments.
> > >
> > > Regarding the total number of instances generated, we created 500 instances for each task-language/language-pair combination, so a total of 8000 = 500 * 4 + 500 * 11 + 500 (i.e., 4 languages for LLM general capabilities, 11 language pairs for MT, 1 language for VLM general capabilities). We realize this is not clear in the paper, so we will add it. Furthermore, we will release all this data upon publication. Thank you for raising this.
> > >
> > > We agree that categorizing data is good to better understand what capabilities the benchmarks are evaluating. However, while ZSB and Arena-Hard have different distributions we do not believe this makes them harder to compare. Both are benchmarks that correlate strongly with human rankings for general capabilities, just as MMLU-Pro, for example, which is very different in nature. In fact, we argue that their complementarity is beneficial in that it enables practitioners to build more comprehensive evaluation suites. In this respect, ZSB has the advantage of allowing the simple creation of different task distributions by adapting the metaprompt (e.g., we could prompt the data generator for more coding-related instances). All in all, beyond their distributions and nature, it is crucial that benchmarks correlate strongly with human rankings and that it is clear what exactly they are evaluating. Thus, we are thankful that you raised this issue and prompted us to run this analysis.
> > >
> > > For the complexity analysis, we used Llama-3.3-Instruct because we found its analysis accurate upon inspection of a few samples, and we could deploy it quickly for the purpose of the discussion period. However, we are happy to replicate the analysis with a stronger/more recent model for the camera-ready version of the paper.
> > >
> > > Regarding lower resourced languages, we will add the by-LP results to the paper, as we mentioned in our first response. However, it is not possible to do the same experiments for LLM General Capabilities, since the human rankings we leveraged—from the [Chatbot Arena](https://lmarena.ai/leaderboard)—both currently and at time of writing, only contain per-language information for English, Chinese, French, German, Spanish, Russian, Japanese, and Korean. In fact, we initially included Korean because it is arguably the lowest-resourced language of this group. Thus, while it would be simple to create an LLM general capabilities benchmark for a lower-resourced language with ZSB, we still have no way of validating its effectiveness because of the lack of human rankings. Creating such rankings is an interesting direction for future work.
> > >
> > > We hope these clarifications address your remaining concerns.

---

> ### Author Response · Authors · 2025-06-01
> **Response (1/2)**
>
> Thank you for your review. We address your comments and questions below.
>
> >The generated data are not analyzed or explained in any depth. The paper does not tell the reader what kind of prompts were created, how well the generated references complete these tasks, or provide any statistic other than the number of examples generated per task (500). Aside from translation abilities, it is unclear what abilities this methodology really evaluates.
> [...]
> What are the tasks generated and what are their basic statistics? [...]
>
> Thank you for pointing this out. In Tables 5 to 17 we detail the distribution of meta-prompt attribute values, providing some insight into dataset variety. Also, in line 81, we state that, with our meta-prompt template, it is possible to generate 9832320 unique data instances. We also provide the meta prompts used for data generation in Appendix A, given some insight on the type of data that will be generated, and what it is meant to evaluate.
>
> To further address this issue, we have classified all our LLM general capabilities instances in terms of complexity, and placed them into 10 topic categories. We also do this with the ArenaHard data, to put our datasets in perspective with human-derived data. Please refer to our official comment for more details on this. We will include this analysis in the final version of the paper.
>
> Furthermore, we will release our datasets upon publication, facilitating further analysis and scrutiny by the community. Please let us know if this alleviates your concern.
>
> >The methodology has several limiting requirements: 1) strong instruction-following model capable of generating fluent text, 2) a strong judge LLM capable of giving meaningful direct assessments, 3) tasks that can be assessed meaningfully via direct assessment. These requirements mean that for many languages and tasks, this methodology is likely to perform poorer than reported here. It would strengthen the paper to include experiments on an under-resourced language or structured prediction task to determine if this methodology is able to deal with adverse conditions.
>
> Respectfully, we believe these requirements are not very limiting. More and more models that satisfy 1) and 2) are becoming widely available and cheaper to run (examples are the Gemma3 and Qwen3 family of models). Indeed, one of the main advantages of ZSB is precisely that it is model agnostic. As open models improve and become more accessible, so will ZSB benchmarks become better and cheaper to create. For example, since the time of writing, in less than 6 months, a wide variety of extremely capable open models have come out (e.g., Qwen3, Gemma-3), so it is reasonable to expect that requirements 1) and 2) will be even easier to fulfill as time goes by.
>
> Requirement 3) is not a hard requirement of ZSB. The framework requires the creation of a judgment prompt, but this need not be DA. In fact, we show in Table 4 that different evaluation methodologies (e.g., pairwise comparison) can be used to achieve similar pairwise accuracy. We chose to frame the tasks in a DA fashion because it was a simple yet effective approach (see Table 4), but extending the framework to other evaluation settings and rubrics requires only adapting the judgment prompt. We will make this clearer in the final version.
>
> Finally, we agree that it is relevant to evaluate ZSB on under-resourced languages. In fact, we included lower-resourced language pairs (LPs) in our machine translation evaluation suite: English-Icelandic and English Ukrainian. Interestingly, ZSB outperforms other MT metrics on both cases (1.0 vs 0.83 PA, and 0.83 vs 0.66, respectively). We realize these LP-specific results were not included in our submission, and we will include them in the final version.

---

### Official Review · Reviewer_B5mp · 2025-05-12

**Rating:** 4
**Confidence:** 4
**Ethics Flag:** 1

**Summary:**

The paper presents zero-shot-benchmarking, a novel framework designed to create benchmarks fully automatically using language models.This framework leverages language models for both synthetic test data generation and evaluation. This is achieved by creating a meta prompt. This meta prompt instructs a language model  to create test instances for a specific task. To ensure variety in the generated data, the meta prompt incorporates placeholder attributes (like topic, style, difficulty, length, etc.) The authors found that variety in these attributes is important for benchmark quality. The meta prompt can also optionally include a request for a reference answer or be accompanied by external data like images.


ZSB requires minimal human effort, needing only two prompts: a meta prompt for data generation and a judgment prompt for evaluation. This design makes it a flexible and scalable framework for end-to-end automatic benchmarking, applicable for any task.  The system rankings derived from ZSB correlate strongly with gold (human) ranking.

**Questions To Authors:**

The authors may explore the Llmits and dependencies of LLM judges- the framework relies fundamentally on LLMs acting as evaluators. While the paper demonstrates that open models can be strong judges, especially larger ones, a deeper discussion or analysis of the limitations of LLM judges (e.g., their potential blind spots, susceptibility to adversarial attacks on responses, or how task complexity affects their performance) would strengthen the paper.

Address biases in synthetic data more comprehensively-  the paper acknowledges that synthetic data may encode biases present in the models used to create it. While safety scores are included for transparency, they are explicitly stated as "not a complete solution". Strengthening the paper could involve exploring more rigorous methods for detecting and mitigating biases within the generated datasets, perhaps through explicit fairness metrics or analysis of how different generator models introduce different types of biases.

**Reasons To Accept:**

The paper proposed
(a)  a novel framework for creating high-quality benchmarks fully automatically by leveraging language models for both synthetic test data creation and evaluation
(b) The framework is designed to require as little human effort as the creation of two prompts
(c) Unlike previous works that primarily used Pairwise Comparison (PWC), the authors specifically opted for Direct Assessment (DA) with a 6-point scale in their judgment prompt

**Reasons To Reject:**

Diversity: The authors have not reported how diverse  the datasets are

Inheriting and exaggerating bias: The LLM may have biases on certain topics, those biases will likely be inherited and potentially exaggerated in the synthetic data. This can lead to models trained or evaluated on such benchmarks to perpetuate or even amplify those biases in real-world deployment

Overfitting to synthetic data: Models optimized or benchmarked against synthetic data might "overfit" to the specific characteristics, patterns, or even flaws of the generated data.

Authors mentioned this technique to be used as to complement the human curated dataset, to remove some bias. then this approach will not be a zero-shot generation process.

---

> ### Author Response · Authors · 2025-06-01
> **Response (2/2)**
>
> >Diversity: The authors have not reported how diverse the datasets are
>
> Thank you for pointing this out. In Tables 5 to 17 we detail the distribution of meta-prompt attribute values, providing some insight into dataset variety. Also, in line 81, we state that, with our meta-prompt template, it is possible to generate 9,832,320 unique data instances. We also provide the meta prompts used for data generation in Appendix A, giving some insight on the type of data that will be generated, and what it is meant to evaluate.
>
> To further address this issue, we have classified all our LLM general capabilities instances in terms of complexity, and placed them into 10 topic categories (as discussed in our first official comment at the top of the forum). We also do this with the ArenaHard data, to put our datasets in perspective with human-derived data. Please refer to our official comment for more details on this. We will include this analysis in the final version of the paper.
>
> Furthermore, we will release our datasets and all prompts (meta-prompts and judge prompts) upon publication, facilitating further analysis and scrutiny by the community.
>
> >Authors mentioned this technique to be used as to complement the human curated dataset, to remove some bias. then this approach will not be a zero-shot generation process.
>
> As mentioned in our Ethics statement, we believe all automatic evaluation should ultimately complement, rather than substitute, human evaluation. By design, ZSB is a zero-shot process. However, its flexibility is such that it is possible to integrate other signals during data creation, including human ones. We see this flexibility as an advantage of the framework, not a limitation.
>
>
> >The authors may explore the limits and dependencies of LLM judges- the framework relies fundamentally on LLMs acting as evaluators. While the paper demonstrates that open models can be strong judges, especially larger ones, a deeper discussion or analysis of the limitations of LLM judges (e.g., their potential blind spots, susceptibility to adversarial attacks on responses, or how task complexity affects their performance) would strengthen the paper.
>
> As we mention in our Introduction and Related Work sections, LLM-as-a-judge is not a novel paradigm, but rather one that has been increasingly popular because of its effectiveness. As such, there is a considerable body of recent work on the limitations and biases of LLM Judges [4,5,6,7]. Thus, we respectfully point out that studying their limitations would likely overlap significantly with existing literature, and would not be central to our work, whose fundamental premise goes way beyond the step of evaluation. We show that ZSB can be used to create benchmarks that strongly correlate with human rankings by mostly using an LLM judge “out-of-the-box”.
>
> We believe to have addressed all your main questions, please let us know if any further clarification is necessary. In case our answer alleviates your concerns, we kindly ask you to consider updating your recommendation score.
>
> [4] Thakur, A. S., Choudhary, K., Ramayapally, V. S., Vaidyanathan, S., & Hupkes, D. (2024). Judging the judges: Evaluating alignment and vulnerabilities in llms-as-judges. arXiv preprint arXiv:2406.12624.
>
> [5] Li, D., Jiang, B., Huang, L., Beigi, A., Zhao, C., Tan, Z., ... & Liu, H. (2024). From generation to judgment: Opportunities and challenges of llm-as-a-judge. arXiv preprint arXiv:2411.16594.
>
> [6] Szymanski, A., Ziems, N., Eicher-Miller, H. A., Li, T. J. J., Jiang, M., & Metoyer, R. A. (2025, March). Limitations of the llm-as-a-judge approach for evaluating llm outputs in expert knowledge tasks. In Proceedings of the 30th International Conference on Intelligent User Interfaces (pp. 952-966).
>
> [7] Verga, P., Hofstatter, S., Althammer, S., Su, Y., Piktus, A., Arkhangorodsky, A., ... & Lewis, P. (2024). Replacing judges with juries: Evaluating LLM generations with a panel of diverse models. arXiv preprint arXiv:2404.18796.

---

> ### Author Response · Authors · 2025-06-01
> **Response (1/2)**
>
> Thank you for your review. We address your comments and questions below.
>
> >Inheriting and exaggerating bias: The LLM may have biases on certain topics, those biases will likely be inherited and potentially exaggerated in the synthetic data. This can lead to models trained or evaluated on such benchmarks to perpetuate or even amplify those biases in real-world deployment
>
> >Overfitting to synthetic data: Models optimized or benchmarked against synthetic data might "overfit" to the specific characteristics, patterns, or even flaws of the generated data.
>
> >Address biases in synthetic data more comprehensively- the paper acknowledges that synthetic data may encode biases present in the models used to create it. While safety scores are included for transparency, they are explicitly stated as "not a complete solution". Strengthening the paper could involve exploring more rigorous methods for detecting and mitigating biases within the generated datasets, perhaps through explicit fairness metrics or analysis of how different generator models introduce different types of biases.
> We respond here similarly to our response to reviewer D9Dp.
>
> We want to highlight that we are aware of these risks, and we openly discuss them in our Ethics statement. Furthermore, as we also discuss in our official comment above, we produce an additional column with the complexity of each instance generated, and we split the data into 10 topic categories to provide further insight on what exactly our datasets are evaluating. We also do this with the ArenaHard data, to put our datasets in perspective with human-derived data. We will include this analysis in the final version of the paper.
>
> It should be noted that humans are not free of biases, and, as such, methods that rely exclusively on human-annotated data are not either. An excellent example are the state-of-the-art machine translation evaluation metrics we report in Table 2. They are trained on large amounts of human data, and yet there is a large body of work extensively studying their biases [1,2,3] (e.g., gender bias, self-preference bias). In our work, we provide safety scores to mitigate some of them, and, now, motivated by reviewers’ comments, we additionally obtain complexity scores, and categorize our data into more insightful buckets (see our official comment for details on this). Furthermore, at time of publication, the datasets will be made available for further exploration by the community, along with the metaprompts that were used to generate every single instance of the ZSB benchmarks.
>
> A more complete solution to address these issues, including the exploration of methods for detecting and mitigating biases through explicit fairness metrics (as the reviewer suggests) is a very interesting direction for future work, though the breadth of the topic deserves a paper on its own—such an endeavour lies outside the scope of this paper. We see ZSB as a “live” benchmark which facilitates this kind of research.
>
> [1] Stanovsky, G., Smith, N. A., & Zettlemoyer, L. (2019, July). Evaluating Gender Bias in Machine Translation. In Proceedings of the 57th Annual Meeting of the Association for Computational Linguistics (pp. 1679-1684).
>
> [2] Zouhar, V., Chen, P., Lam, T. K., Moghe, N., & Haddow, B. (2024, November). Pitfalls and Outlooks in Using COMET. In Proceedings of the Ninth Conference on Machine Translation (pp. 1272-1288).
>
> [3] Pombal, J., Guerreiro, N. M., Rei, R., & Martins, A. F. (2025). Adding chocolate to mint: Mitigating metric interference in machine translation. arXiv preprint arXiv:2503.08327.

---

### Official Review · Reviewer_D9Dp · 2025-05-12

**Rating:** 5
**Confidence:** 4
**Ethics Flag:** 1

**Summary:**

This paper introduces "Zero-shot Benchmarking" (ZSB), a proposed framework for the automated generation of benchmarks for language models. The core concept involves utilizing language models for both synthetic test data creation and subsequent evaluation, ostensibly requiring only two prompts. The authors aim to address persistent challenges in LM evaluation: the high cost and rapid obsolescence of human-annotated datasets, and the difficulty of developing robust task-specific metrics.

ZSB is presented as simple, flexible, scalable (to new tasks and languages), and model-agnostic. To demonstrate its efficacy, the authors create benchmarks for several text-based tasks (general capabilities in multiple languages, translation) and a multimodal task (vision-language in English). They proceed to rank numerous models, reporting that ZSB-derived rankings exhibit strong correlation with human judgments, in some cases surpassing standard benchmarks. Further ablation studies indicate that while open models can be used, performance is influenced by judge model size and dataset variety.

**Questions To Authors:**

* The ablation study indicates that judge models may perform better on data generated by models from the same family. Does this suggest a potential "self-preference" or "model-family" bias, and what are the implications for benchmark fairness when evaluating a diverse range of models?
* Given the identified importance of dataset variety, what strategies beyond increasing meta-prompt attributes were considered or could be integrated into ZSB to enhance diversity and challenge? For instance, could external knowledge sources or adversarial generation techniques be incorporated?
* Regarding the Direct Assessment methodology: How sensitive are the results to the specific phrasing and definitions within the 6-point Likert scale rubric? Was any form of judge model calibration undertaken to ensure consistent application of the scale?

**Reasons To Accept:**

- The paper addresses a significant and timely problem. The challenge of evaluating rapidly advancing language models is widely recognized, and existing methods face considerable limitations. The motivation for a new approach is well-articulated.
- The central proposal—employing LMs for both data generation and evaluation within an integrated framework—is conceptually sound. While the constituent elements (LM-as-a-judge, synthetic data generation) are not novel, their unification into an adaptable system applicable across diverse tasks and languages represents a small contribution to the literature.
- The ZSB framework is described with reasonable clarity. The comprehensive experimental validation across multiple tasks, languages, and modalities is a strength. The ablation studies provide valuable insights into the factors influencing ZSB performance.
- The paper is generally well-written and accessible. The main arguments are conveyed effectively, and the use of figures aids in understanding the proposed framework and results.
- The potential impact of ZSB, particularly its claimed low cost, could be substantial for democratizing benchmark creation, especially for under-resourced languages or novel tasks. The framework's capacity for benchmarks to evolve in tandem with LM improvements is an appealing feature. The commitment to release code and data will facilitate broader adoption and further research.

**Reasons To Reject:**

- A significant concern is the reliance on a large-scale, proprietary model (Claude-Sonnet-3.5) for optimal performance in both data generation and judgment, as demonstrated in the primary experiments. While ablations suggest viability with open models, the observed performance degradation is considerable.
- The framework's complete reliance on LM-generated data and LM-based judgments introduces inherent risks. The quality of synthetic data—its fidelity to real-world complexities, potential inheritance of biases from the generator LM, and effective diversity—is paramount. While safety scores are provided, a more thorough analysis of the qualitative aspects of generated instances versus human-derived data would strengthen the paper's claims.
- I question the consistency and reliability of Direct Assessment (DA) using a 6-point Likert scale. The paper could benefit from a more detailed investigation into the sensitivity of DA results to rubric phrasing and judge model calibration.
- In the VLM benchmark generation, the framework utilized pre-existing images paired with LM-generated text prompts, rather than fully synthetic image generation. While the authors acknowledge synthetic image creation as future work, this current implementation somewhat limits the "zero-shot" aspect for the visual modality.
- The use of M-ArenaHard (queries derived from the Chatbot Arena, the source of the gold-standard human rankings) evaluated using the ZSB judging framework as an "upper bound" warrants careful interpretation. While intended to isolate the impact of test data provenance, the shared origin of M-ArenaHard queries with the gold standard inherently advantages this baseline, potentially making comparisons with ZSB less direct than implied.
- The claim of task-agnosticism may be challenged by the practicalities of prompt engineering. Developing effective meta-prompts for data generation and judgment prompts for evaluation, particularly for entirely novel or highly specialized tasks, could still represent a substantial engineering effort, potentially limiting true "zero-shot" applicability in all scenarios.

---

> ### Author Response · Authors · 2025-06-01
> **Response 3/3**
>
> >The claim of task-agnosticism may be challenged by the practicalities of prompt engineering. Developing effective meta-prompts for data generation and judgment prompts for evaluation, particularly for entirely novel or highly specialized tasks, could still represent a substantial engineering effort, potentially limiting true "zero-shot" applicability in all scenarios.
>
> We understand your point. What we mean with “task-agnostic” is that ZSB is not tied to a particular task, but allows flexible incorporation of diverse tasks with minimal effort. Indeed, we evaluate ZSB on a suite of tasks covering different languages and modalities. In total, we cover 3 tasks, 4 languages, 11 language pairs, and 2 modalities. As discussed in Section 3.2, we choose these particular tasks because of their importance in LLM development, their diversity, and the availability of reliable gold rankings. It would be ideal to experiment with novel and highly specialized tasks, but it is hard to obtain gold rankings. We believe we made significant effort to support our claim of task-agnosticism, and encourage the reviewer to re-consider their assessment.
>
> In any case, we respectfully disagree with the claim that prompt engineering effort limits the applicability of our framework. We state, when describing the framework in Section 2 and Figure 1, that the only human effort ZSB requires is the creation of two prompts: one for data generation and another for evaluation. Some tasks, regardless of their novelty and specialization, may require more complex prompts, depending on their specific requirements. This is expected within the framework. Moreover, as illustrated in Figure 1, ZSB is receptive to additional inputs that can further guide the data generation process and may be particularly helpful for novel or highly specialized tasks (e.g., human-annotated examples, knowledge bases, etc.).
>
> >The ablation study indicates that judge models may perform better on data generated by models from the same family. Does this suggest a potential "self-preference" or "model-family" bias, and what are the implications for benchmark fairness when evaluating a diverse range of models?
>
> We believe there may have been a misunderstanding of this ablation, which we will make sure to clarify in the final version of the paper.
> ZSB has three stages where models come up: **A)** the data generation phase; **B)** the response phase; **C)** the evaluation phase, where responses from models used in stage B on the data created in stage A are evaluated.
>
> There is no self-preference concern when using the same models for phases A and C, which is what we explore in our ablations. In fact, we find that there seems to be some synergy in using the same models or models of the same family for both data generation and judging.
>
> Issues of benchmark fairness may arise when using the same models for stages B and C, for example. Indeed, this has been the focus of previous work that we mention in line 101 [4,5]; in fact, some solutions proposed in these works (e.g., using more than one model for judging) can be seamlessly integrated into ZSB. This topic, however, is not directly related to our ablations.
> Please let us know if this explanation clears up the matter.
>
> [4] Verga, P., Hofstatter, S., Althammer, S., Su, Y., Piktus, A., Arkhangorodsky, A., ... & Lewis, P. (2024). Replacing judges with juries: Evaluating LLM generations with a panel of diverse models. arXiv preprint arXiv:2404.18796.
>
> [5] Zhao, R., Zhang, W., Chia, Y. K., Xu, W., Zhao, D., & Bing, L. (2024). Auto-Arena: Automating LLM Evaluations with Agent Peer Battles and Committee Discussions. arXiv preprint arXiv:2405.20267.
>
>
> >Given the identified importance of dataset variety, what strategies beyond increasing meta-prompt attributes were considered or could be integrated into ZSB to enhance diversity and challenge? For instance, could external knowledge sources or adversarial generation techniques be incorporated?
>
> As we point out in Figure 1, yes, external sources of data (e.g., pre-existing data, images) can be used to enhance ZSB. In this work, we chose to focus on extensively ablating components of the meta-prompt itself, because it is a central part of the framework, and where it is easiest to make adjustments. We believe it is appropriate to leave the study of other enhancements to future work.
>
> We believe to have addressed all your main questions, please let us know if any further clarification is necessary. In case our answer alleviates your concerns, we kindly ask you to consider updating your recommendation score.

---

> ### Author Response · Authors · 2025-06-01
> **Response 2/3**
>
> >I question the consistency and reliability of Direct Assessment (DA) using a 6-point Likert scale. The paper could benefit from a more detailed investigation into the sensitivity of DA results to rubric phrasing and judge model calibration.
> [...]
> Regarding the Direct Assessment methodology: How sensitive are the results to the specific phrasing and definitions within the 6-point Likert scale rubric? Was any form of judge model calibration undertaken to ensure consistent application of the scale?
>
> Thank you for pointing this out. We used DA for all tasks considered, and performance was strong, which is, in itself, a testament to the consistency and reliability of DA. However, we agree that it is interesting to assess the sensitivity of the method to rubric phrasing. Thus, we obtain results for 2 additional rubric rephrasings of our judge prompt for the LLM general capabilities benchmarks. The results are the following:
>
> |**Across 3 rubrics**|**English**|**Chinese**|**French**|**Korean**|
> |-|-|-|-|-|
> |Average|0.8470|0.8341|0.8283|0.8419|
> |Standard deviation|0.0090|0.0025|0.0175|0.0074|
>
> The standard deviation of PA is very low in general (in French it is a bit higher), indicating DA is robust to rubric rephrasing. We will include these experiments in the final version of the paper.
>
> >In the VLM benchmark generation, the framework utilized pre-existing images paired with LM-generated text prompts, rather than fully synthetic image generation. While the authors acknowledge synthetic image creation as future work, this current implementation somewhat limits the "zero-shot" aspect for the visual modality.
>
> We respectfully disagree that using existing images for the VLM benchmark is a practical limitation, due to the wide availability of images on the internet. The burden of collecting a small set of images is virtually zero, and can be viewed as part of the process of creating the data generation meta-prompt. Creating synthetic images, on the other hand, is potentially more burdensome, as it would require the creation of one additional prompt (the one for image generation). We also want to highlight that our framework is not tied to fully synthetic data generation—in fact, additional (including human-driven/-annotated/-collected) inputs such as external knowledge bases, benchmark examples, images, etc. can easily complement the data generation process (as illustrated in Figure 1).
>
> >The use of M-ArenaHard (queries derived from the Chatbot Arena, the source of the gold-standard human rankings) evaluated using the ZSB judging framework as an "upper bound" warrants careful interpretation. While intended to isolate the impact of test data provenance, the shared origin of M-ArenaHard queries with the gold standard inherently advantages this baseline, potentially making comparisons with ZSB less direct than implied.
>
> We are aware of the inherent advantage of M-ArenaHard, and chose it precisely because it is an extremely challenging baseline. An upper bound is useful because it puts the performance of our method in perspective, and gives insight into how much we can still improve.
>
> We respectfully disagree that M-ArenaHard and ZSB are not comparable. They are two benchmarks for general capabilities, just as ZSB is comparable with standard general capabilities benchmarks like, MMLU-Pro, or MUSR, even though they are very different in nature. These baselines are used for the same purpose: to rank systems automatically and measure progress. We are not comparing specific components of each method, but rather their effectiveness for automatic evaluation.

---

> ### Author Response · Authors · 2025-06-01
> **Response 1/3**
>
> Thank you for your review. We respond to your comments and questions below.
>
> >A significant concern is the reliance on a large-scale, proprietary model (Claude-Sonnet-3.5) for optimal performance in both data generation and judgment, as demonstrated in the primary experiments. While ablations suggest viability with open models, the observed performance degradation is considerable.
>
> We show that **using Qwen2.5-72B for both data generation and evaluation outperforms standard benchmarks (averaging), the most popular baseline considered, across all languages** on the LLM General Capabilities task (this can be concluded from Tables 1, 3, 24, 25, and 27; see also our response to reviewer hRX1, and our official comment). **The claim that observed performance degradation is considerable with open-models is unfair**. ZSB produces benchmarks that are highly correlated with human judgments—higher even than the most widely used baselines—with an open model
>
> Furthermore, and perhaps more importantly, as we elaborate in our response to reviewer hRX1 and in our second official comment, ZSB is a model-agnostic framework, so it does not rely on any particular type of model (proprietary, or otherwise). If anything, our experiments with open models show that it is possible to build strong benchmarks with open models (see L.228).
>
> As open models improve, so will the performance of ZSB. Indeed, we ran Llama 4 Maverick—an open frontier model not available at time of writing—as a judge on our English LLM General Capabilities benchmark and obtained 0.844 PA. This outperforms the LLama-3.3 model we used, and is much closer to Claude. We are happy to include these experiments, and experiments with more frontier open models to better illustrate the point that ZSB is model agnostic and can improve over time.
>
> >The framework's complete reliance on LM-generated data and LM-based judgments introduces inherent risks. The quality of synthetic data—its fidelity to real-world complexities, potential inheritance of biases from the generator LM, and effective diversity—is paramount. While safety scores are provided, a more thorough analysis of the qualitative aspects of generated instances versus human-derived data would strengthen the paper's claims.
>
> We want to highlight that we are aware of these risks, and we openly discuss them in our Ethics statement. Furthermore, as we also discuss in our official comment, we produce an additional column with the complexity of each instance generated, and we split the data into 10 topic categories to provide further insight on what exactly our datasets are evaluating. We also do this with the ArenaHard data, to put our datasets in perspective with human-derived data. We will include this analysis in the final version of the paper.
>
> It should be noted that humans are not free of biases, and, as such, methods that rely exclusively on human-annotated data are not either. An excellent example are the state-of-the-art machine translation evaluation metrics we report in Table 2. They are trained on large amounts of human data, and yet there is a large body of work extensively studying their biases [1,2,3] (e.g., gender bias, self-preference bias). In our work, we provide safety scores to mitigate some of them, and, now, motivated by reviewers’ comments, we additionally obtain complexity scores, and categorize our data into more insightful buckets (see our official comment for details on this). Furthermore, at time of publication, the datasets will be made available for further exploration by the community, along with the metaprompts that were used to generate every single instance of the ZSB benchmarks. A more complete solution to address these issues, including the exploration of methods for detecting and mitigating biases through explicit fairness metrics is a very interesting direction for future work, though the breadth of the topic deserves a paper on its own—such an endeavour lies outside the scope of this paper. We see ZSB as a “live” benchmark which facilitates this kind of research.
>
> [1] Stanovsky, G., Smith, N. A., & Zettlemoyer, L. (2019, July). Evaluating Gender Bias in Machine Translation. In Proceedings of the 57th Annual Meeting of the Association for Computational Linguistics (pp. 1679-1684).
>
> [2] Zouhar, V., Chen, P., Lam, T. K., Moghe, N., & Haddow, B. (2024, November). Pitfalls and Outlooks in Using COMET. In Proceedings of the Ninth Conference on Machine Translation (pp. 1272-1288).
>
> [3] Pombal, J., Guerreiro, N. M., Rei, R., & Martins, A. F. (2025). Adding chocolate to mint: Mitigating metric interference in machine translation. arXiv preprint arXiv:2503.08327.

---

### Official Review · Reviewer_hRX1 · 2025-05-13

**Rating:** 6
**Confidence:** 4
**Ethics Flag:** 1

**Summary:**

This paper presents "Zero-shot Benchmarking" (ZSB), a framework for creating high-quality benchmarks to evaluate language models. It shows how ZSB uses language models for both test data creation and evaluation with minimal human effort. This paper requires only two prompts: a meta prompt for data generation and a judgment prompt for evaluation. This paper demonstrates ZSB's flexibility across different tasks and languages, creating benchmarks for general capabilities in four languages, translation across eleven language pairs, and vision-language capabilities. Experimental results shows ZSB rankings correlate strongly with human rankings, often outperforming standard benchmarks. In ablation studies, the paper finds that open models can create good benchmarks, judge model size affects performance, and dataset variety is more important than size. This paper aims to make benchmark creation more accessible, especially for specialized tasks where collecting real-world data would be costly or impractical.

**Questions To Authors:**

### Questions
* Lines 181-187: M-ArenaHard is regarded as an upper bound, but
* Line: 183: What is "ArenaHardAuto" mentioned here? Is it different from automatically translated ArenaHard by Google Translate?
  * After reading couple more lines, I realized that it's [1]. Can you add the citation here too?
* Figure 2: Is Claude Sonnet 3.5 the model used to generate the test instances here? If so, what would happen with the open-weight models? What would be the cost implication of it?
### Suggestions

[1] Tianle Li, Wei-Lin Chiang, Evan Frick, Lisa Dunlap, Tianhao Wu, Banghua Zhu, Joseph E Gonzalez, and Ion Stoica. From crowdsourced data to high-quality benchmarks: Arena505 hard and benchbuilder pipeline. arXiv preprint arXiv:2406.11939, 2024c.

**Reasons To Accept:**

* The main takeaway raises an important question on what should the benchmark dataset if it can be generated by an LLM.
* Evaluation across various models with different sizes and families including vision-language models

**Reasons To Reject:**

* There's no clear generalizable guidance on which model to choose for zero-shot benchmarking for open weight models (Lines 238-239)
  * The pairwise accuracy can large differ across different languages/datasets (Table 22)
* The analysis on using open weight models for data generation is limited since the main model used to generate data is Claude Sonnet 3.5
* The "LLM General Capabilites" which is the main characteristic that trying to measure through zero-shot benchmarking is defined as the human preference of models on the Chatbot Arena, which itself is questionable that it measures "LLM General Capabilities" or not.

---

> ### Author Response · Authors · 2025-06-01
> **Response**
>
> Thank you for your review. We will address your comments and questions below.
>
> >There's no clear generalizable guidance on which model to choose for zero-shot benchmarking for open weight models (Lines 238-239). The pairwise accuracy can large differ across different languages/datasets (Table 22)
>
> Thanks for asking this question. In fact, one of the main advantages of our framework is that ZSB is agnostic to the model used, which allows for the quality of benchmarks to improve as existing models improve.
>
> Nevertheless, to illustrate this point, we ran Llama-4-Maverick as a judge (an open frontier model not available at time of writing) on our English LLM General Capabilities dataset, and obtained 0.84 PA, which vastly outperforms the Llama-3.3 model we reported in the paper, and is competitive with Claude.
>
> Thus, the guideline for choosing an open model is to experiment with the strongest models available at any given time (that fit compute restrictions), and choose the one whose generated data/evaluations fit best the requirements of the task of interest. Similar reasoning applies to closed models. We can make this more clear in the final version of the paper.
>
> At the time of writing, we wanted to assess whether using existing open models would work well for both data generation and evaluation. Remarkably, when analyzing Tables 1, 3, 24, 25, and 27, despite the variance across languages you mention, using Qwen2.5-72B-Instruct for both ZSB steps (bottom right corner of Tables 3, 24, 25, and 27) outperforms standard benchmarks (averaging) across all languages. However, even if it did not, and perhaps most importantly, the merit of ZSB as a model-agnostic framework would not be compromised.
>
> We are happy to run the rest of our benchmarks with Llama 4, and other recent open models, and include those experiments in the final version of the paper.
>
> We also address this issue in our official comment at the top.
>
> We are not entirely sure how your mention to Table 22 relates to this issue. The table  shows the performance of individual standard benchmarks used as baselines (not our method). We will make this more clear in the final version.
>
> >The analysis on using open weight models for data generation is limited since the main model used to generate data is Claude Sonnet 3.5
> [..]
> Figure 2: Is Claude Sonnet 3.5 the model used to generate the test instances here? If so, what would happen with the open-weight models? What would be the cost implication of it?
>
> We understand this point, and, for the final version of the paper, we will produce a version of Figure 2 when using Qwen2.5-72B-Instruct as a data generator and judge—the conclusions are similar to those when using Claude. Thank you for raising this.
> As far as cost is concerned, it depends; if using an API like Together AI, the cost of using a model like Qwen2.5-72B is similar to Claude-3.5. If, on the other hand, one has compute to run such a model locally, the cost is virtually zero (apart from the cost of the compute itself).
>
> >After reading couple more lines, I realized that it's [1]. Can you add the citation here too?
>
> Yes, thank you for pointing this out. We will add the citation in the final version of the paper.
>
> Please let us know if any remaining clarification is needed.

---

> ### Comment · Reviewer_hRX1 · 2025-06-09
>
> I would like to thank the authors for the detailed reply.
>
> >  In fact, one of the main advantages of our framework is that ZSB is agnostic to the model used, which allows for the quality of benchmarks to improve as existing models improve.
>
> My main point is that 1) the claim needs to be backed up, and 2) claiming "model agnostic" needs to be careful and more granular. Lines 239-240 "choosing the right model is crucial for the quality of the benchmark—for example, Qwen is especially strong in Korean, while Llama lags behind." already has a sentence describing it so I would suggest rather than claiming to be "model agnostic", the pairwise accuracy is model dependent (which is obvious).
>
> > we ran Llama-4-Maverick as a judge
> I would not recommend using Llama-4 in the further experiments due to the issue highlighted in [1] and an example output in [2]. According to Section 3.2, "LLM General Capability" is defined as the preference on The Chatbot Arena (Zheng et al., 2023) dataset, and Llama-4 has the doubts to explicitly trained to be optimized for the Chatbot Arena outputs (e.g., more emojis). I believe this is worth noting as discussion or limitations somewhere in the submission.
>
> [1] https://x.com/lmarena_ai/status/1909397817434816562
>
> [2] https://huggingface.co/spaces/lmarena-ai/Llama-4-Maverick-03-26-Experimental_battles

---

> ### Author Response · Authors · 2025-06-10
> **Response**
>
> Thank you for your additional feedback.
>
> > My main point is that 1) the claim needs to be backed up, and 2) claiming "model agnostic" needs to be careful and more granular. Lines 239-240 "choosing the right model is crucial for the quality of the benchmark—for example, Qwen is especially strong in Korean, while Llama lags behind." already has a sentence describing it so I would suggest rather than claiming to be "model agnostic", the pairwise accuracy is model dependent (which is obvious).
>
> We agree that the quality of a given benchmark (as measured, for example, by pairwise accuracy) is model-dependent. Our point about model agnosticity is that ZSB, conceptually, does not depend on any particular kind of model (e.g., proprietary). This is true by the definition of the framework, but we also show that it is true in practice by experimenting with various models—strong results can be obtained with open models (e.g., ZSB with Qwen2.5-72B outperforms standard benchmark averaging). We will carefully revise how we introduce this claim in the paper, taking into account your feedback.
>
> > I would not recommend using Llama-4 in the further experiments due to the issue highlighted in [1] and an example output in [2]
>
> Thank you for raising this issue. We are happy to run further experiments with other state-of-the art open LLMs, such as Qwen3 and Gemma-3. These additional experiments will also serve to reinforce what we mean by ZSB being model-agnostic. We are now too close to the end of the discussion period to obtain results in time, but there is time to obtain them for the camera-ready version. Regarding including some discussion about the shortcomings of the Arena and Llama 4 in the final version of the paper, we completely agree. This information had not come out at time of writing, but it is definitely relevant.
>
> Please let us know if you have any further concerns.

---

### Author Response · Authors · 2025-06-01
**Official comment addressing reviewer concerns (2/2)**

**Concern #2: Reliance on proprietary models in the experimental section / concerns on how to use open models for zero-shot benchmarking.** (raised by reviewers hRX1, D9Dp)

Zero-shot Benchmarking (ZSB) is a model-agnostic framework. It does not rely on any particular type of model. This is a central attribute of the framework and one of its main advantages.

In the paper, we tested several models (proprietary/open, large/small, Qwen/Llama families) to assess how good ZSB benchmarks could be in different practical settings. We believe that no part of this analysis put the framework in jeopardy, but rather showcased its flexibility in practice. While using Claude yielded the best benchmarks overall, **using Qwen2.5-72B-Instruct still outperforms popular standard benchmarks across all languages** (see the bottom right corner of Tables 3, 24, 25, and 27, and compare the values with Table 1). Thus, **ZSB can be used to create SotA benchmarks with open models**.

Even if that were not the case, **the model-agnosticity of ZSB enables users to create improved benchmarks as better models come out**; this is one of the main advantages of the framework. So, if open models had not performed well at the time of writing—which they did—better ZSB benchmarks could be made once stronger models came out. Indeed, since the time of this submission, several new SotA model families have been released (e.g., Qwen3, Gemma-3, Llama 4). We believe it is reasonable to expect that more and better models will be released in the future, raising the potential of ZSB benchmarks. To illustrate this point, we ran Llama 4 Maverick as a judge on our English LLM General capabilities benchmark, achieving 0.844 PA. **This means Llama 4 Maverick greatly surpasses the Llama version we report in the paper (~0.75 PA), and is competitive with Claude (0.85 PA)**.

We are happy to make this property of the framework clearer in the final version of the paper, and to include experiments with more recent open models (e.g., Llama 4 and Qwen3) for all our datasets. We hope this comment clarifies these concerns, and we encourage the reviewers to revisit the paper and their assessments with it in mind.

---

### Author Response · Authors · 2025-06-01
**Official comment addressing reviewer concerns (1/2)**

We are glad that the reviewers found ZSB to be a novel and timely contribution due to its flexibility, effectiveness and low cost (reviewers D9Dp, B5mp), and that they found our analysis and evaluation to be varied and thorough (reviewers hRX1, PEQ1).

In these official comments, we address two concerns raised by more than one reviewer here to facilitate dialogue during the discussion period.

**Concern #1: Lack of a qualitative analysis on the created benchmarks, e.g., on the diversity of the datasets, and the capabilities they evaluate.** (raised by reviewers D9Dp, B5mp, PEQ1)

In the paper, we included distributions of all the meta-prompt attributes for every task (see Tables 5 to 17 in Appendix A.2), as well as all meta prompts (Appendix A.1) and judge prompts (Appendix A.3), which provide information on what the benchmarks are supposed to evaluate. We also state that it is possible to generate 9,832,320 unique instances with our meta prompts (L. 81). Furthermore, we will release the data and all prompts (meta-prompts and judge prompts) upon publication, enabling further scrutiny and analysis from the community.

Nevertheless, we thank reviewers for raising these concerns, as we recognize that doing more qualitative analysis will help understanding the benchmarks better and will strengthen our work. Thus, using our English LLM General Capabilities benchmark as a testbed (500 instances), we took two concrete actions:
1) We classify all data instances into 10 categories: coding, advice and brainstorming, question answering, mathematical reasoning, creative writing and persona, summarization, text correction or rewriting, classification, translation, and other. These categories are meant to represent coarse-grained tasks for which LLMs are used [1,2], providing a better general understanding of the capabilities our benchmarks evaluate.
2) Similarly to our safety evaluation, we add a new column to our data, rating every instance in terms of the degree of reasoning required to answer from 1 to 5. This should give a notion of how challenging the benchmark is.

We use LLama-3.3-70B to run this analysis automatically, and repeat it for Arena-Hard so as to have a point of comparison with human-written data. The results are as follows:

|**Category**|**ZSB (ours)**|**Arena-Hard (human-written)**|
|-|-|-|
|Coding|0|253|
|Advice and Brainstorming|218|91|
|Question Answering|69|72|
|Mathematical Reasoning|9|41|
|Creative Writing and Persona|202|27|
|Summarization|0|5|
|Text Correction or Rewriting|0|5|
|Classification|0|3|
|Translation|0|3|
|Other|2|0|

Remarkably, while Arena-Hard is mostly a benchmark for code, ZSB has many more instances of advice and brainstorming, and creative writing. Question answering is the third most prevalent category for both types of data. This demonstrates that, while there is some complementarity between our benchmarks, their principal focus is on different capabilities: ZSB on writing, Arena-Hard on coding.

In terms of complexity:

|**Degree of reasoning required to answer**|**ZSB (ours)**|**Arena-Hard (human-written)**|
|-|-|-|
|5 (highest)|228|36|
|4|245|314|
|3|10|29|
|2|17|120|
|Average|4.37|3.53|

ZSB benchmarks seem to have heavier reasoning requirements, with most instances being assigned scores of 5 to 4.

We hope this analysis also addresses some of the reviewers’ concerns regarding taking more actions to describe and avoid bias to facilitate safe use of our benchmarks. Crucially, note that users of ZSB can choose to create more instances of certain categories, safety, or reasoning levels at will by simply adapting the data generation meta prompt. The on-the-fly creation of benchmarks tailored to specific needs is possible with ZSB.

In the final version of the paper, we are happy to expand this analysis to the rest of our benchmarks, and we will release all prompts and code used.

[1] Li, T., Chiang, W. L., Frick, E., Dunlap, L., Wu, T., Zhu, B., ... & Stoica, I. (2024). From Crowdsourced Data to High-Quality Benchmarks: Arena-Hard and BenchBuilder Pipeline. CoRR.

[2] Handa, K., Tamkin, A., McCain, M., Huang, S., Durmus, E., Heck, S., ... & Ganguli, D. (2025). Which economic tasks are performed with ai? evidence from millions of claude conversations. arXiv preprint arXiv:2503.04761.

---

### Author Response · Authors · 2025-06-09
**Comment for Area Chairs**

Dear Area Chairs,

While we did our best to address the concerns of reviewers hRX1, D9Dp, B5mp, they have yet to engage in the discussion period. We urge the reviewers to provide feedback before the end of the discussion period. In any case, in this comment we provide a brief summary of our responses to their concerns.

Regarding the concerns raised by reviewers D9Dp and B5mp about qualitative analysis of our datasets: we wrote an official comment presenting new experiments where we categorize our data into specific tasks and complexity levels, comparing it to Arena-Hard, a human-written dataset. This brings new and useful insight on what exactly our benchmarks are evaluating. Furthermore, we commit to performing the same analysis on the rest of our data for the camera-ready version of the paper.

Regarding the concern of reliance on proprietary models raised by reviewers hRX1 and D9Dp: we also wrote an official comment describing how ZSB is model-agnostic and how, even still, strong results that outperform widely-used standard benchmarks can be obtained using solely open models. We also ran an experiment with a more recent open model (Llama-4 Maverick) that highlights how ZSB can be improved by using better models. We commit to running this experiments for the rest of our data for the camera-ready version of the paper. We reinforce that the model-agnosticity of ZSB is one of its main advantages, allowing better benchmarks to be created as stronger open models are released.

Furthermore, we address specific concerns in our responses to each of the reviewers.

We believe we have sufficiently clarified most concerns, and that the new experiments we presented significantly strengthen the paper. We urge the Area Chairs to consider this new information when reviewing the paper, despite the lack of engagement of the reviewers during the discussion period.

Thank you for your attention.

---

### Decision · Program_Chairs · 2025-07-08

**Decision:**

Accept

**Comment:**

This paper introduces Zero-shot Benchmarking (ZSB), a flexible framework for benchmark creation that leverages LLMs for both data generation and evaluation. The core idea—automated, prompt-based benchmarking—is timely and impactful, and the paper presents extensive empirical validation across multiple languages, tasks, and modalities. While the reliance on proprietary models (Claude) and limited qualitative analysis raised concerns among reviewers, the authors responded with new experiments, additional analysis, and a strong rebuttal. Overall, reviewer scores are mixed (4, 5, 5, 6, 7), but given the conceptual novelty, practical utility, and demonstrated improvements over standard baselines, I recommend Accept.